# Engineered PW$_{12}$-polyoxometalate docked Fe sites on CoFe hydroxide anode for durable seawater electrolysis

Xun He [1,2,7], Yongchao Yao[1,3,7], Min Zhang[4], Yilei Zhou[5], Limei Zhang[3], Yuchun Ren[2], Kai Dong[4], Hong Tang[2], Jue Nan[2], Xingli Zhou[1], Han Luo [3], Binwu Ying[3], Qi Yu [5] ✉, Fengming Luo[1] ✉, Bo Tang [4,6] ✉ & Xuping Sun [1,4] ✉

Seawater electrolysis driven by offshore renewable energy is a promising avenue for large-scale hydrogen production but faces challenges in designing robust anodes that suppress surface chlorine reactions and corrosion at high current densities. Here we report a strategy by selectively docking PW$_{12}$-polyoxometalate (PW$_{12}$-POM) onto Fe sites of CoFe hydroxide anode to modulate the electronic structure of adjacent Co active centers and regulate Cl$^-$/OH$^-$ adsorption for efficient alkaline seawater oxidation. Our CoFe-based anode achieves low overpotentials, high catalytic selectivity, and notable durability, with continuous operation at 1 A cm$^{-2}$ for over 1300 hours and at 2 A cm$^{-2}$ more than 600 hours. Theoretical calculations and ex situ/in situ analyses reveal that PW$_{12}$-POM coordination at Fe sites stabilizes Fe, suppresses its leaching, modulates Co acidity, promotes OH$^-$ adsorption, and protects metal sites from Cl$^-$ corrosion.

Hydrogen, as a clean and versatile energy carrier, is essential for reducing $CO_2$ emissions and transitioning to a sustainable energy system[1–5]. Water electrolysis, as a key technology for energy storage, offers a viable pathway for $H_2$ production[6–14]. However, conventional electrolyzers depend on highly purified water, limiting their application in offshore wind farms and coastal photovoltaic plants[15,16]. Direct seawater electrolysis could harness Earth's abundant water resources and advance the water–energy nexus[17–24], but the presence of high chloride concentration (~ 0.5 M) in seawater presents major challenges. Chloride ions accelerate electrode degradation through chemical corrosion (Supplementary Note 1) and electrochemical chloride oxidation reaction (ClOR) pathways, generating aggressive species like $Cl_2$ and $ClO^-$ that compete with oxygen evolution reaction (OER) and damage catalysts[25–32]. Alkalinizing seawater can reduce the ClOR thermodynamic potential gap with OER by 480 mV, helping suppress

side reactions[33–35], yet sustaining ampere-level densities remains difficult due to persistent corrosion and increased internal resistance.

Noble-metal-based anodes exhibit strong OER activity and corrosion resistance in Cl$^-$-rich environments, but their scarcity and cost drive the exploration of earth-abundant alternatives[36–40]. Recently, transition-metal-based compounds (e.g., Fe, Co, Ni sulfides, phosphides, nitrides) have emerged as promising alkaline seawater oxidation (ASO) anodes, as these materials can generate protective, anion-rich surfaces (e.g., $SO_4^{2-}$, $PO_4^{3-}$, $NO_3^-$) to repel Cl$^-$ via electrostatic interactions[41–45]. Strategies, such as engineered electrolytes[46–48], physical barrier layers (e.g., $MoO_3$, $MnO_2$)[49,50], or surface chloride-immobilizing layers (e.g., AgCl, IrCl$_x$)[39,40], have also been employed to limit Cl$^-$ access. Despite these advances and operation beyond 1000 hours have been demonstrated, many of these approaches inadvertently compromise the adsorption of crucial oxygen

[1]Center for High Altitude Medicine, West China Hospital, Sichuan University, Chengdu, Sichuan, China. [2]Institute of Fundamental and Frontier Sciences, University of Electronic Science and Technology of China, Chengdu, Sichuan, China. [3]Department of Laboratory Medicine/Clinical Laboratory Medicine Research Center, West China Hospital, Sichuan University, Chengdu, Sichuan, China. [4]College of Chemistry, Chemical Engineering and Materials Science, Shandong Normal University, Jinan, Shandong, China. [5]School of Materials Science and Engineering, and Shaanxi Laboratory of Catalysis, Shaanxi University of Technology, Hanzhong, Shaanxi, China. [6]Laoshan Laboratory, Qingdao, Shandong, China. [7]These authors contributed equally: Xun He, Yongchao Yao. ✉e-mail: qiyu@snut.edu.cn; luofengming@wchscu.edu.cn; tangb@sdnu.edu.cn; xpsun@uestc.edu.cn

intermediates, thereby limiting catalytic efficiency. Moreover, at high overpotentials or current densities, the current reported $Cl^-$-repelling strategies generally lose their effectiveness due to increased $Cl^-$ adsorption tendencies, triggering accelerated metal leaching, electrode degradation, and ultimately causing electrolysis failure[35,51,52]. In addition, the anion-rich surfaces, though effective at repelling $Cl^-$, may also repel $OH^-$, hindering selective $OH^-$ adsorption[27,33,45]. Thus, developing robust strategies capable of simultaneously boosting catalytic activity and mitigating chlorine-induced degradation during ASO is a pressing challenge.

In this work, an engineered $PW_{12}$-polyoxometalate ($PW_{12}$-POM)-docked CoFe hydroxide anode achieves prolonged operation of 1300 hours at $1 A cm^{-2}$ and 600 hours at $2 A cm^{-2}$. Detailed ex situ/ in situ analyses coupled with theoretical calculations identify that $PW_{12}$-POM docking on Fe sites stabilizes Fe, modulates electronic states of neighboring Co active sites, enhances $OH^-$ adsorption selectivity, and prevents $Cl^-$-induced corrosion. Notably, the anode exhibits high gas selectivity, producing negligible amounts of chlorine by-products even under high current densities. In addition, the CoFe-based anode demonstrates notable durability in membrane electrode assembly (MEA) electrolyzers, maintaining stability for 1000 hours at $1 A cm^{-2}$, indicating its potential for practical applications.

## Results

### Engineering of CoFe hydroxide with $PW_{12}$-POM

The engineered $PW_{12}$-POM-docked CoFe layered double hydroxide on Ni foam ($PW_{12}$-CoFe LDH/NF) anode was fabricated by submerging hydrothermally grown CoFe LDH/NF in an aqueous $PW_{12}$-POM solution. X-ray diffraction (XRD) patterns of CoFe LDH/NF and $PW_{12}$-CoFe LDH/NF exhibit identical Bragg reflections in alignment with the hexagonal-phase LDH (PDF#50-0235, Fig. 1a). With no shift observed at 11.7° for the (003) plane for $PW_{12}$-CoFe LDH/NF compared to CoFe LDH/NF, it is confirmed that $PW_{12}$-POM incorporation does not alter the interlayer spacing, indicating surface binding rather than intercalation. Scanning electron microscopy (SEM) images show that CoFe LDH nanosheets are aligned perpendicularly, with $PW_{12}$-POM incorporation leaving their morphology unchanged (Supplementary Fig. 1 and Fig. 1b). Elemental mapping images and energy-dispersive X-ray (EDX) spectrum analysis verify the existence of Co, Fe, P, W, and O throughout the NF (Supplementary Fig. 2). Inductively coupled plasma-optical emission spectrometry (ICP-OES) analysis further demonstrates a P/W ratio of ~1:13.1, closely consistent with $PW_{12}$-POM stoichiometry, and $PW_{12}$-POM accounts for 36.6 wt% of $PW_{12}$-CoFe LDH (Supplementary Table 1). The atomic force microscopy image confirms that the $PW_{12}$-CoFe LDH nanosheet is ~3 nm thick (Supplementary Fig. 3). The ultrathin, vertically oriented array on NF ensures active-site exposure to the electrolyte, thereby boosting its catalytic activity. Transmission electron microscopy (TEM) image of $PW_{12}$-CoFe LDH further displays the orderly stacked ultrathin nanosheets (Fig. 1c). Aberration-corrected scanning transmission electron microscopy (AC-STEM) images (Supplementary Fig. 4 and Fig. 1d) indicate the presence of nanoclusters on the CoFe LDH (012) plane. Notably, the structural details observed from the edge of a single nanocluster match the configuration of a $PW_{12}$-POM molecule (Fig. 1e, f). STEM image and its corresponding EDX elemental mapping images in Fig. 1g further confirm the uniform distribution of Co, Fe, O, W, and P. X-ray photoelectron spectroscopy (XPS) spectra in the Co 2*p* and Fe 2*p* regions reveals that the introduction of $PW_{12}$-POM induces slight shifts in Co 2*p* and marked shifts in Fe 2*p* toward higher binding energies, as well as shifts toward lower binding energies in the W 4*f* and P 2*p* regions (Supplementary Fig. 5). In comparison, $PW_{12}$-Co(OH)$_2$ exhibits negligible signals in the W 4*f* region (Supplementary Fig. 6). Collectively, these results demonstrate selective coordination of $PW_{12}$-POM to Fe sites and electron transfer from CoFe LDH to $PW_{12}$-POM. The Raman spectra shown in Supplementary Fig. 7 show the signals of $PW_{12}$-POM,

particularly with the intensified terminal $W = O$ asymmetric stretching vibration relative to the $W = O$ symmetric stretching vibration, indicative of robust coordination between $PW_{12}$-POM and CoFe LDH. X-ray absorption near-edge structure (XANES) spectroscopy was applied to investigate changes in the coordination environment and oxidation states of Co and Fe within $PW_{12}$-CoFe LDH. Compared with CoFe LDH, the Fe K-edge of $PW_{12}$-CoFe LDH shifts to higher energy (Fig. 1h), while the Co K-edge exhibits only a slight shift (Fig. 1i). In contrast, the W $L_3$-edge XANES spectrum shifts toward lower energy relative to pristine $PW_{12}$-POM (Supplementary Fig. 8), aligning with the XPS results. Further estimation of Fe and Co oxidation states was performed using first derivative analysis and fitted average oxidation states from the Fe and Co K-edge XANES spectra (Fig. 1j and Supplementary Fig. 9). As shown in Supplementary Fig. 10 and the inset of Fig. 1j, the average oxidation states of Fe and Co in $PW_{12}$-CoFe LDH are 3.51 and 2.34, respectively, higher than those in CoFe LDH (2.62 and 2.13). Extended X-ray absorption fine structure (EXAFS) analysis of the Fe K-edge reveals an increase in the relative intensity of the Fe−O bond at 1.5 Å in $PW_{12}$-CoFe LDH compared to CoFe LDH (Fig. 1k), suggesting enhanced Fe−O coordination due to oxygen from $PW_{12}$-POM binding to Fe. In addition, the Fe−O−Co bond at 2.5 Å exhibits a positive shift and stretching, likely indicating the formation of Fe−O−W interfacial bonds, with the peak shift implying strong interactions at the $PW_{12}$-CoFe LDH interface. Co K-edge EXAFS spectra reveal almost no change in Co−O coordination, with only a slight extension of Co−O−Fe bonds caused by Fe−O−W bond formation (Supplementary Fig. 11). In addition, the pronounced alteration of the W−O bond further supports Fe−O−W coordination (Supplementary Fig. 12). Pronounced variations in the Fe K-edge EXAFS oscillation function, compared to the minimal changes in the Co K-edge, suggest distinct local atomic arrangements for Fe in $PW_{12}$-CoFe LDH (Supplementary Figs. 13 and 14). Wavelet transform EXAFS analysis of the Fe and Co K-edges further supports these results (Fig. 1l, m and Supplementary Figs. 15 and 16).

### Electrochemical tests

Among the tested anodes, $PW_{12}$-CoFe LDH/NF shows enhanced water oxidation activity in 1 M KOH relative to CoFe LDH/NF, $RuO_2$/NF, and NF (Fig. 2a). As anticipated, the introduction of $PW_{12}$-POM improves the intrinsic activity of CoFe LDH/NF (Supplementary Figs. 17 and 18). The $PW_{12}$-CoFe LDH/NF electrode was further evaluated in both alkaline simulated seawater and alkaline seawater, and its activity shows only slight attenuation relative to its activity in 1 M KOH (Supplementary Fig. 19). Notably, $PW_{12}$-CoFe LDH/NF achieves current densities of 100, 500, and 1000 mA cm$^{-2}$ at overpotentials of 265, 325, and 368 mV, respectively, outperforming CoFe LDH/NF, which requires 285, 372, and 458 mV to attain the same benchmarks (Supplementary Fig. 20). In addition, we evaluated alternative POM substitutions via the same synthesis method, including $SiW_{12}$-CoFe LDH/NF and $PMo_{12}$-CoFe LDH/NF, but neither substitution produced improvements in OER activity observed with $PW_{12}$-CoFe LDH/NF (Supplementary Figs. 21 and 22). Remarkably, $PW_{12}$-CoFe LDH/NF delivers comparable ASO activity than previously reported CoFe-based and other representative ASO anodes (Supplementary Table 2). After normalizing by the electrochemically active surface area (Supplementary Figs. 23 and 24), $PW_{12}$-CoFe LDH/NF still outperforms CoFe LDH/NF at high current densities. Electrochemical impedance spectroscopy tests performed from 1.42 to 1.52 V confirms the much lower charge transfer resistance for the $PW_{12}$-CoFe LDH/NF electrode compared to CoFe LDH/NF (Supplementary Fig. 25). In situ Bode phase plots further indicate that the transition peaks in the $10^{-1}$–$10^1$ Hz frequency range differ between these electrodes. The smaller phase angle of $PW_{12}$-CoFe LDH/NF implies more charges are engaged in Faradaic processes at the interface[53,54]. Notably, the phase angle of $PW_{12}$-CoFe LDH/NF decreases most rapidly, suggesting easier polarization and faster reaction kinetics (Fig. 2b). Under an industrial current density of $1.0 A cm^{-2}$,

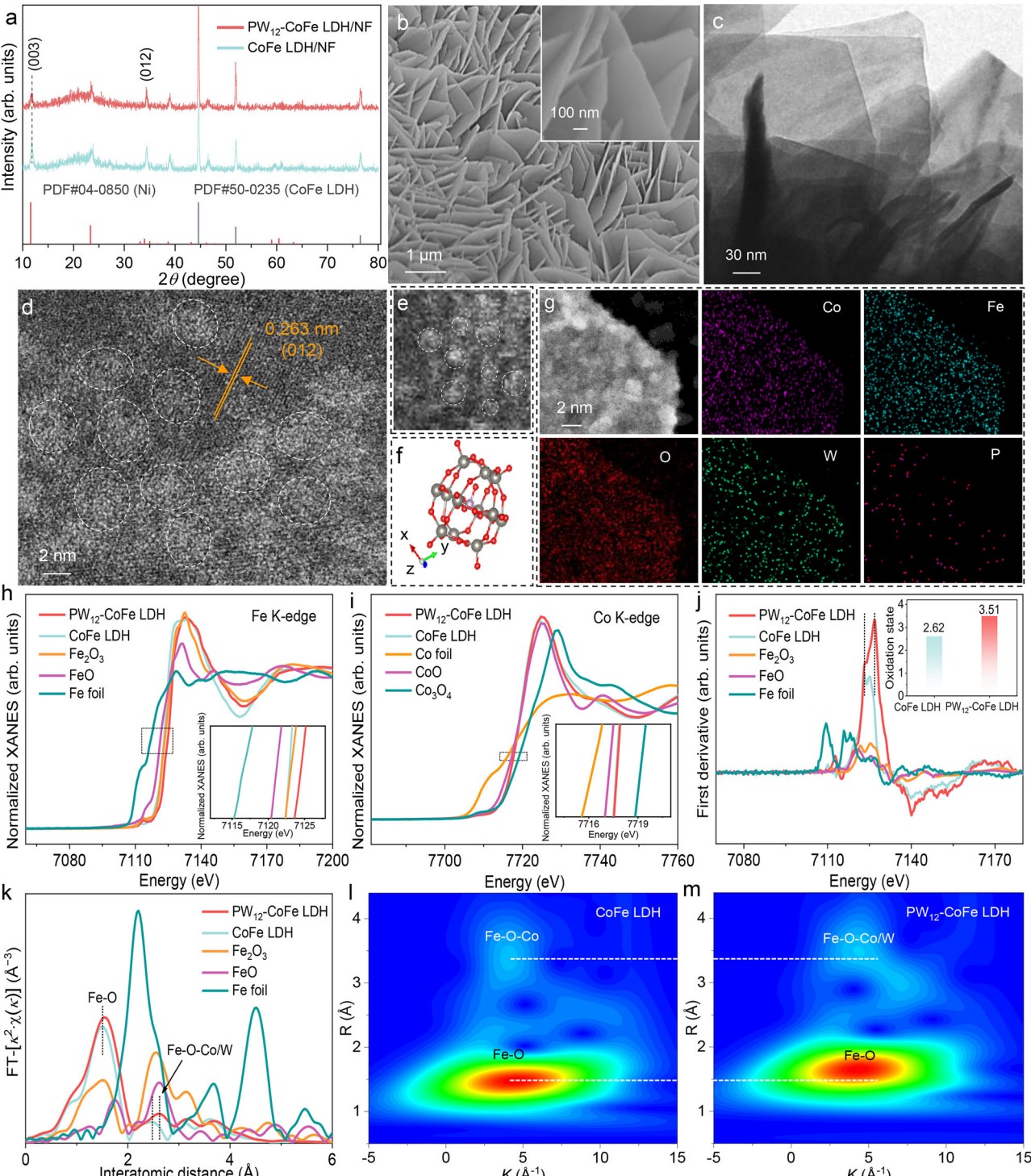

**Fig. 1 | Material characterizations. a** XRD patterns of CoFe LDH/NF and PW₁₂-CoFe LDH/NF. **b** SEM images of PW₁₂-CoFe LDH/NF. **c** TEM image of PW₁₂-CoFe LDH. **d** High-magnification AC-STEM image of PW₁₂-CoFe LDH. **e** Extracted AC-TEM image from (**d**) for detailed analysis. **f** Rotated view of a PW₁₂-POM molecule in a ball-and-stick model format. Gray, red, and purple spheres represent W, O, and P atoms, respectively. **g** STEM image and its corresponding elemental mapping images of PW₁₂-CoFe LDH. **h** Normalized Fe K-edge XANES spectra of PW₁₂-CoFe LDH, CoFe LDH, Fe₂O₃, FeO, and Fe foil. **i** Normalized Co K-edge XANES spectra of

PW₁₂-CoFe LDH, CoFe LDH, Co₃O₄, CoO, and Co foil. **j** First derivative Fe K-edge spectra of PW₁₂-CoFe LDH, CoFe LDH, Fe₂O₃, FeO, and Fe foil. The inset shows the oxidation states of Fe for PW₁₂-CoFe LDH and CoFe LDH. **k** FT-EXAFS spectra of Fe K-edge for PW₁₂-CoFe LDH, CoFe LDH, Fe₂O₃, FeO, and Fe foil. Wavelet transform of k²-weighted EXAFS signals for (**l**) CoFe LDH and (**m**) PW₁₂-CoFe LDH. R represents the distance from the central atom, and K is the wavelength of the oscillation. Source data are provided as a Source Data file.

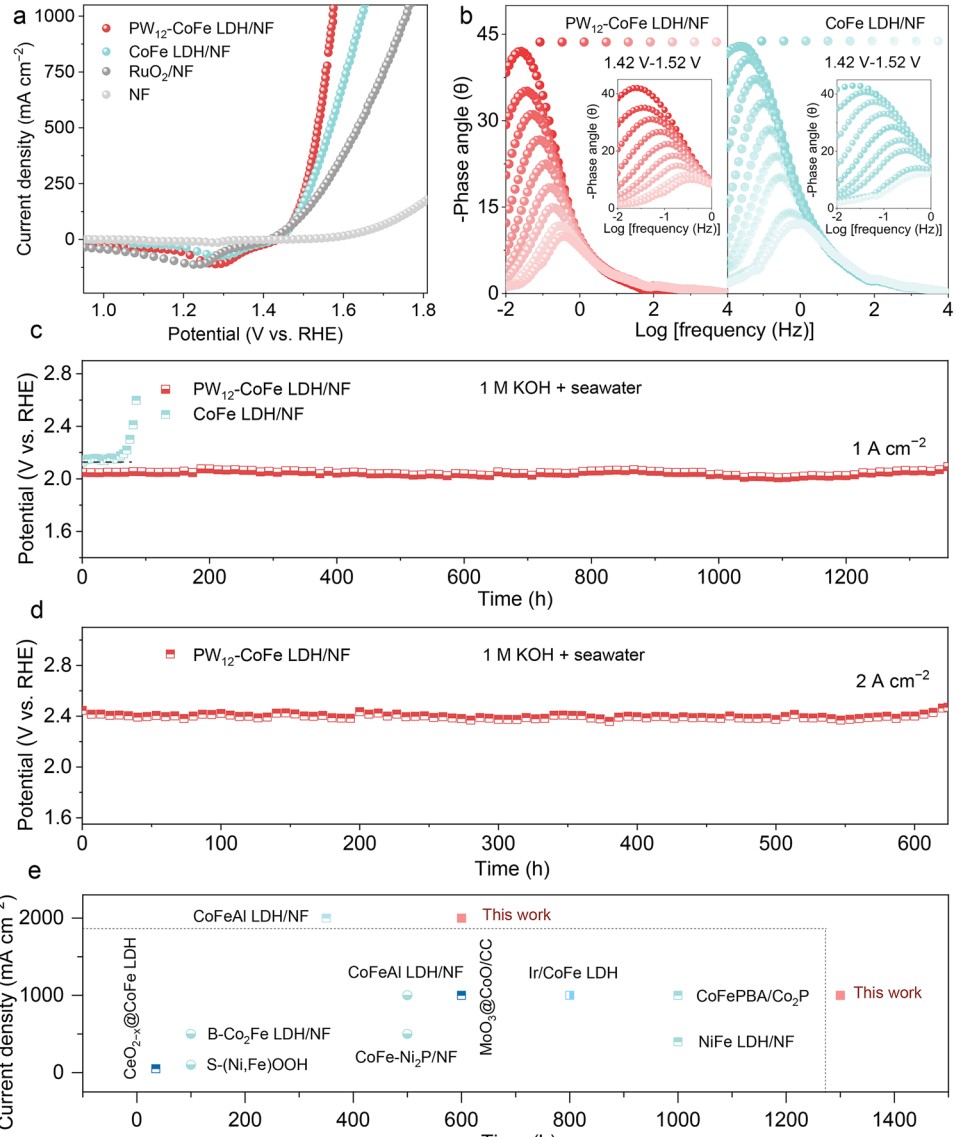

**Fig. 2 | Electrochemical performance evaluation. a** Evaluation of catalytic activities at a scan rate of 5 mV s$^{-1}$ with 100% *iR* correction. **b** Bode plots for CoFe LDH/NF and PW$_{12}$-CoFe LDH/NF at different potentials. Inset shows a magnified view of the low-frequency region. **c** Comparison of chronopotentiometry curves of CoFe LDH/NF and PW$_{12}$-CoFe LDH/NF at 1 A cm$^{-2}$ without *iR* correction. **d** Chronopotentiometry curve of PW$_{12}$-CoFe LDH/NF at 2 A cm$^{-2}$ without *iR* correction. **e** Comparison of stability at 1 and 2 A cm$^{-2}$ of PW$_{12}$-CoFe LDH/NF with recently reported ASO anodes. Source data are provided as a Source Data file.

PW$_{12}$-CoFe LDH/NF remains stable for over 1300 hours, whereas CoFe LDH/NF deactivates after 60 hours (Fig. 2c). After 1300 hours of electrolysis at 1 A cm$^{-2}$, ultraviolet−visible spectroscopy detected only trace active chlorine in the PW$_{12}$-CoFe LDH/NF electrolyte, compared with markedly high levels for CoFe LDH/NF after 60 hours (Supplementary Fig. 26). In addition, PW$_{12}$-CoFe LDH/NF achieves ~99.7% Faradaic efficiency for O$_2$ production, signifying a highly selective four-electron reaction pathway (Supplementary Fig. 27). The observed positive shift in its corrosion potential relative to the PW$_{12}$-POM-free electrode (Supplementary Fig. 28), further indicates its enhanced corrosion resistance. At a higher current density of 2.0 A cm$^{-2}$, PW$_{12}$-CoFe LDH/NF can maintain stable operation for over 600 hours (Fig. 2d). The electrode operates reliably under fluctuating power conditions (1/2 A cm$^{-2}$), making it one of the most stable CoFe-based ASO anodes reported so far (Supplementary Table 3, Fig. 2e). Overall, its performance is comparable to that of other ASO anodes reported in recent studies.

## Investigation of activity enhancement and corrosion resistance

We employed density functional theory (DFT) calculations to investigate the adsorption of PW$_{12}$-POM on CoFe LDH and determine the associated binding energies. The calculations reveal that PW$_{12}$-POM, when adsorbed on Fe sites, exhibits a binding energy of 0.103 eV, considerably lower than the 0.864 eV measured for adsorption on Co sites (Supplementary Fig. 29). This result further indicates a preferential adsorption of PW$_{12}$-POM at Fe sites, which is consistent with our XPS and XANES data. Next, we investigated the role of PW$_{12}$-POM in tuning the electron configurations of CoFe LDH, and the results indicate that when PW$_{12}$-POM bonds to Fe, there is a marked electron loss from Fe along with electron gain by PW$_{12}$-POM (Fig. 3a). This strong interaction further alters the electronic structure of nearby Co atoms, causing partial electron depletion. Electron localization function (ELF) analysis further reveals increased electron delocalization for both Fe and Co (Fig. 3b). The electron redistribution induced by PW$_{12}$-POM coordination can facilitate CoFe LDH catalyst dehydrogenation more

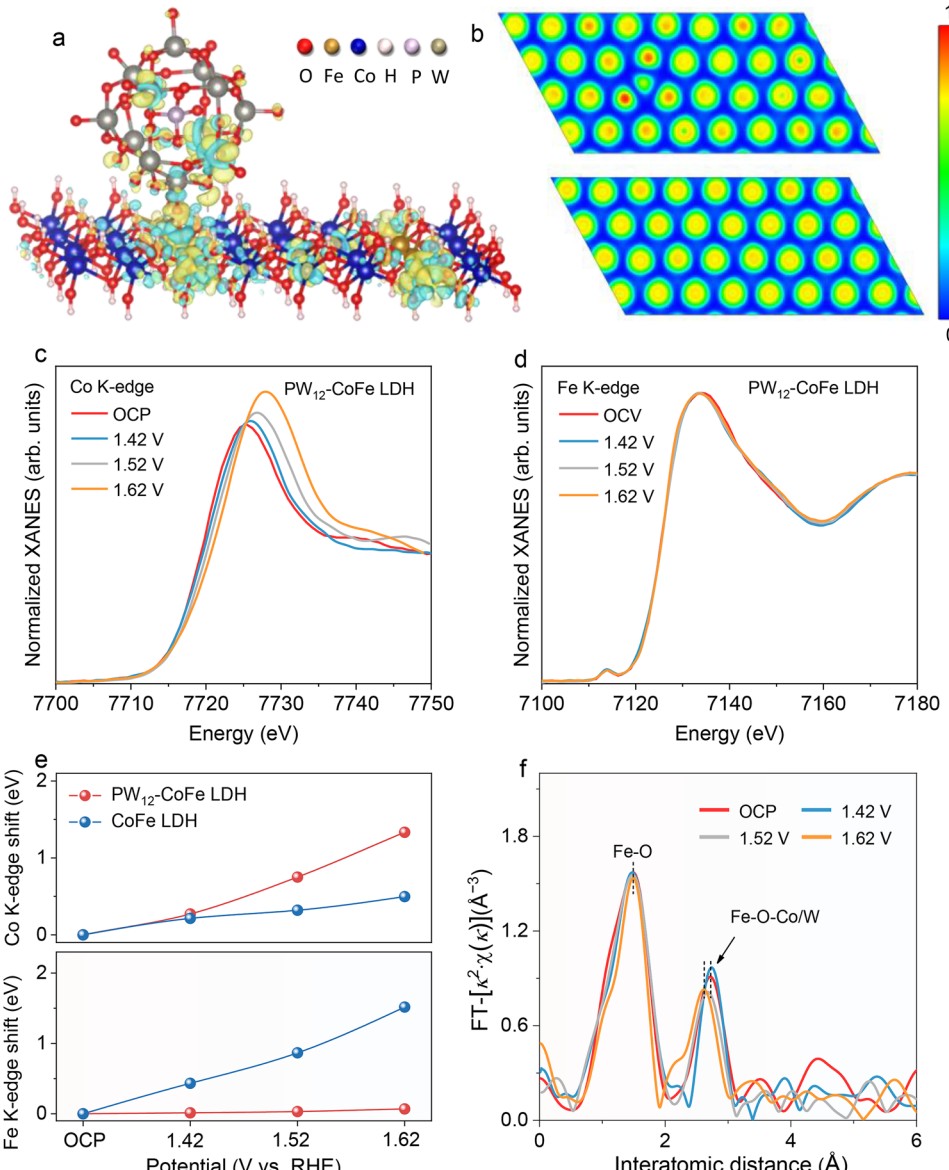

**Fig. 3 | Insights into catalyst structure changes. a** Charge density difference diagram of PW$_{12}$-CoFe LDH. Yellow: charge accumulation; cyan: charge depletion. **b** ELF plot of PW$_{12}$-CoFe LDH (top) and CoFe LDH (bottom). In situ (**c**) Co and (**d**) Fe K-edge XANES spectra of PW$_{12}$-CoFe LDH during ASO. **e** Shifts in the Co and Fe K-edge positions extracted from in situ XANES spectra. **f** In situ FT-EXAFS of Fe K-edge spectra for PW$_{12}$-CoFe LDH. Source data are provided as a Source Data file.

favorable. In contrast, PMo$_{12}$-POM and SiW$_{12}$-POM display smaller energy differences between Fe and Co adsorption on CoFe LDH (SiW$_{12}$-POM: Fe 0.461 eV, Co 0.776 eV; PMo$_{12}$-POM: Fe 0.380 eV, Co 0.816 eV), implying competitive adsorption at both metal centers. When adsorbed onto Co sites, these POMs could induce electron accumulation at Co atoms, consistent with XPS results, thereby hindering dehydrogenation and reducing activity (Supplementary Figs. 30 and 31). In situ Raman spectroscopy was further conducted to gain insights into CoFe LDH dehydrogenation and changes upon PW$_{12}$-POM incorporation (Supplementary Fig. 32). With increasing applied potential, typical metal oxyhydroxides were formed on PW$_{12}$-CoFe LDH/NF, as indicated by the increased intensity ratio of the 452.6 and 526.3 cm$^{-1}$ peaks, along with the appearance of a new peak at 1060 cm$^{-1}$ corresponding to −OOH[48,52,55]. Remarkably, this structural transformation occurs at lower potentials for PW$_{12}$-CoFe LDH/NF than for CoFe LDH/NF, indicating that PW$_{12}$-POM coordination accelerates dehydrogenation and promotes the formation of catalytically favorable metal oxyhydroxides during ASO. XPS analysis of PW$_{12}$-CoFe LDH at different electrolysis

times (0, 1, 20, 40, 60, 80 hours) shows a continuous rise in the Co oxidation state, with over 80% of Co converting to Co$^{3+}$ and almost no change observed for Fe (Supplementary Figs. 33 and 34), which confirms that the strong PW$_{12}$-POM−Fe interaction at the PW$_{12}$-CoFe LDH interface stabilizes Fe.

To further elucidate the catalytic mechanism, we conducted in situ XAS tests from open-circuit potential (OCP) to 1.62 V during ASO for PW$_{12}$-CoFe LDH and CoFe LDH (Fig. 3c, d and Supplementary Fig. 35). The Co K-edge for PW$_{12}$-CoFe LDH exhibits a faster and more pronounced shift to higher energy compared to CoFe LDH (Fig. 3e), indicating accelerated dehydrogenation and oxidation at the Co sites, in line with the operando Raman results. Conversely, the Fe K-edge of PW$_{12}$-CoFe LDH remains nearly unchanged, while a continuous positive shift is observed for CoFe LDH, suggesting that the PW$_{12}$-POM coordination effectively stabilizes the Fe. EXAFS fitting reveals that the Fe−O bond length and coordination number remain nearly constant for PW$_{12}$-CoFe LDH across the applied potentials, indicating that the local coordination environment of Fe is well preserved during ASO

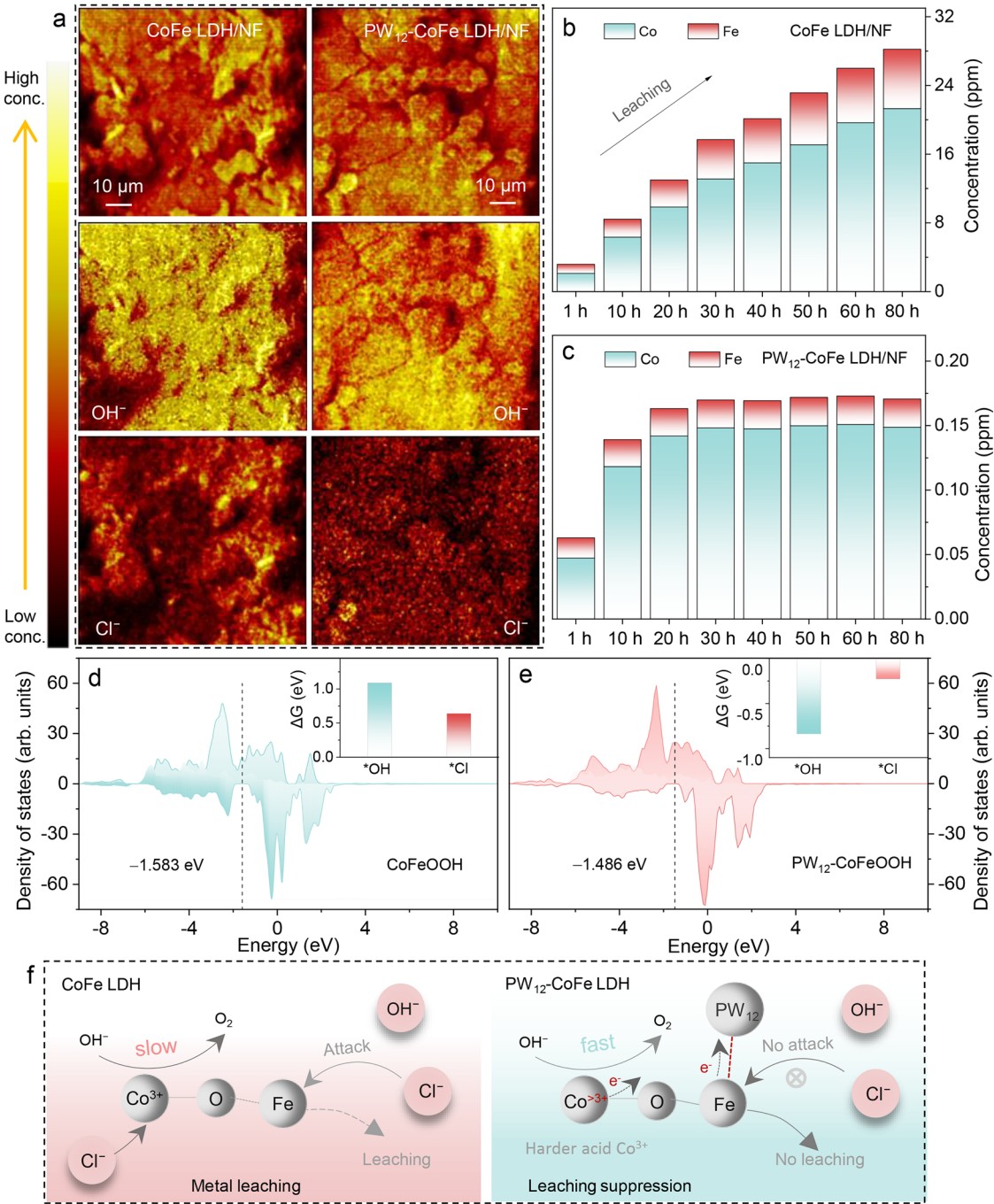

**Fig. 4 | Investigating Cl⁻/OH⁻ adsorption and corrosion resistance. a**TOF-SIMS mapping of the full spectrum, OH⁻ and Cl⁻ fragments on CoFe LDH/NF and PW₁₂-CoFe LDH/NF electrode surface following a 24-hour test. Time-resolved Fe and Co leaching for (**b**) PW₁₂-CoFe LDH/NF and (**c**) CoFe LDH/NF over 80 h. PDOS plots of Co 3 *d* orbital for (**d**) CoFeOOH and (**e**) PW₁₂-CoFeOOH. Insets show the adsorption energies for ˙OH and ˙Cl. **f** Schematic illustration of the PW₁₂-POM-boosted ASO mechanism. Source data are provided as a Source Data file.

(Fig. 3f). In sharp contrast, CoFe LDH displays a noticeable bond elongation and reduced coordination for Fe–O, reflecting structural distortion and Fe dissolution under strongly oxidative, chloride-containing conditions (Supplementary Fig. 36). In addition, the Fe–O–Co/W coordination for PW₁₂-CoFe LDH exhibit a slight shortening in bond distance along with a moderate decrease in coordination number, reflecting a controlled local rearrangement upon Co oxidation. In comparison, CoFe LDH shows irregular changes in the Fe–O–Co coordination, reflecting increasing structural disorder and vulnerability of Fe sites at high anodic potentials amid concentrated OH⁻ and corrosive Cl⁻.

Time-of-flight secondary ion mass spectrometry (TOF-SIMS) was employed to measure OH⁻ and Cl⁻ concentrations on the surface of CoFe LDH/NF and PW₁₂-CoFe LDH/NF electrodes after 24 hours of electrolysis. As shown in Fig. 4a, TOF-SIMS mapping images reveal that both electrodes are predominantly covered with OH⁻. However, the Cl⁻ signal on PW₁₂-CoFe LDH/NF is markedly weaker compared to CoFe LDH/NF, which confirms that introducing PW₁₂-POM effectively reduces Cl⁻ adsorption at the metal sites and repels Cl⁻. Further analysis of the electrolyte over 1 to 80 hours of electrolysis shows a continuous increase for Co and Fe leaching from the CoFe LDH/NF electrode (Supplementary Table 4 and Fig. 4b). In comparison, PW₁₂-CoFe LDH/NF

shows trace leaching for Co, Fe, P, and W within the first 10 hours, and retains compositional stability thereafter (Supplementary Table 5 and Fig. 4c). SEM images obtained after 80 hours of electrolysis show that the NF framework is corroded and fractured and that the CoFe LDH structure collapsing and aggregating (Supplementary Fig. 37a, b). However, the incorporation of $PW_{12}$-POM preserves the NF framework and maintains the structural integrity of the nanosheet array even after 1300 hours of continuous electrolysis (Supplementary Fig. 37c, d), supporting its role in resisting chloride corrosion. In addition, XRD pattern and TEM image from long-term electrolysis confirm that $PW_{12}$-CoFe LDH retains its layered structure (Supplementary Fig. 38). Raman spectrum indicates that $PW_{12}$-POM remains stably present and that most of the CoFe LDH is converted into metal oxyhydroxides (Supplementary Fig. 39a). XPS analysis further shows that $PW_{12}$-POM stabilizes Fe and confirms extensive formation of metal oxyhydroxides (Supplementary Fig. 39b–d).

Given that the $PW_{12}$-POM docking effectively stabilizes Fe and Co as the principal catalytic sites, we carried out DFT calculations on the Co center for both CoFeOOH and $PW_{12}$-CoFeOOH to determine their adsorption energies for ˙OH and ˙Cl intermediates. Using the optimized structures presented in Supplementary Fig. 40, we calculated the free energy changes (ΔG) associated with the adsorption of these species. The results reveal that, upon incorporation of $PW_{12}$-POM, the free energy for ˙OH adsorption markedly decreases from 1.087 eV to − 0.836 eV, while the ΔG for ˙Cl adsorption decreases only slightly to −0.226 eV (Supplementary Fig. 41, insets in Fig. 4d, e). This pronounced difference in adsorption energies (− 0.836 eV versus − 0.226 eV) reveals a strong adsorption selectivity toward ˙OH over ˙Cl at Co site, indicating that $PW_{12}$-POM incorporation can avoid ˙Cl adsorption and protect Co from leaching. In addition, projected density of states (PDOS) analysis of the Co 3$d$ orbitals (Fig. 4d, e) reveals that the d-band center for $PW_{12}$-CoFeOOH shifts to − 1.486 eV, in contrast to − 1.583 eV in CoFeOOH, suggesting that $PW_{12}$-POM incorporation can enhance the binding of ˙OH. Although the introduction of $PW_{12}$-POM markedly reduces the free energy for ˙OH adsorption (− 0.836 eV), further calculations reveal no evident increase in the energy barrier for subsequent OER intermediate conversions (Supplementary Fig. 42). Specifically, the barrier of the rate-determining step (˙O to ˙OOH) for $PW_{12}$-CoFeOOH is 2.425 eV, lower than that of CoOOH (2.487 eV), and substantially below the barrier for Cl⁻ oxidation (˙Cl + Cl⁻ to $Cl_2$, 2.946 eV) (Supplementary Fig. 43), verifying the thermodynamic favorability of OER. Charge density difference analysis further reveals an electron transfer of 0.88 |e| from CoFeOOH to $PW_{12}$-POM (Supplementary Fig. 44), resulting in enhanced acidity at Co sites. This increased acidity aligns with both experimental and theoretical results that $PW_{12}$-POM docking favors selective OH⁻ adsorption over Cl⁻. Moreover, the increased negative charge of $PW_{12}$-POM can further improve its ability to repel Cl⁻.

Based on comprehensive ex situ/in situ analyses and DFT calculations, we propose a $PW_{12}$-POM-boosted ASO mechanism (Fig. 4f). The docking of $PW_{12}$-POM onto Fe sites can help stabilize Fe against corrosion and reduce Fe dissolution. Simultaneously, the coordination modulates electron density at neighboring Co centers during ASO, enhancing their electron transfer ability, increasing selective OH⁻ adsorption, and accelerating $O_2$ evolution kinetics, thereby improving catalytic activity and durability.

### Practical electrolysis applications

To assess its practical application for seawater electrolysis, the MEA electrolyzer with an anion exchange membrane (AEM, PiperION-A60) was constructed, as illustrated in Supplementary Fig. 45. In this assembly, anions and water migrate through the AEM, enabling electron transfer at the $PW_{12}$-CoFe LDH/NF anode, while $H_2/O_2$ and seawater are discharged from the chamber (Fig. 5a).

The assembled MEA electrolyzer, with $PW_{12}$-CoFe LDH/NF as the anode and Pt/C/NF as the cathode shows greater electrocatalytic activity compared with the $RuO_2$/NF||Pt/C/NF benchmark at 60 °C (Fig. 5b). Specifically, $PW_{12}$-CoFe LDH/NF||Pt/C/NF reaches a current density of 0.5 A cm⁻² at a cell voltage of 1.99 V, in contrast to the 2.57 V required by $RuO_2$/NF||Pt/C/NF, and it attains 1.0 A cm⁻² at only 2.39 V. The electrocatalytic activity was further enhanced with an increase in operating temperature to 70 °C and 80 °C, and the electrolyzer shows preliminary stability in temperature-variation tests (60–80 °C) (Supplementary Fig. 46). Long-term stability evaluations at 60 and 80 °C further demonstrate durable performance, with sustained electrolysis for over 1000 hours at 1.0 A cm⁻² in alkaline seawater (Fig. 5c). Overall, the results indicate that $PW_{12}$-CoFe LDH/NF||Pt/C/NF with $PW_{12}$-CoFe LDH/NF anode provides a competitive cell voltage at industrial-level current densities and exhibits prolonged electrolysis stability (Supplementary Table 6, Fig. 5d). Thus, $PW_{12}$-CoFe LDH/NF emerges as a highly promising, stable, and efficient anode for practical seawater electrolysis applications.

## Discussion

The $PW_{12}$-CoFe LDH/NF anode demonstrates comparable activity and durability for ASO, sustaining stability for over 1300 hours at 1.0 A cm⁻² and over 600 hours at 2.0 A cm⁻² in three-electrode tests. The $PW_{12}$-boosted ASO process offers two key benefits: (1) strong, targeted $PW_{12}$-POM coordination with Fe sites shields against Cl⁻ and OH⁻ attack, suppressing Fe leaching; (2) $PW_{12}$-POM induced charge depletion at Co sites strengthens Co acid centers, promoting selective OH⁻ adsorption over Cl⁻. Together, these benefits provide metal sites protection against chloride corrosion, preserving electrode integrity and boosting catalytic activity and durability. In a two-electrode MEA electrolyzer, it demonstrates stable operation over 1000 hours at 1 A cm⁻². This work delivers a highly active and durable anode for alkaline seawater electrolysis and gains in-depth insight into the interaction between Cl⁻-resisting species and catalytic sites for guiding the rational design of ASO catalysts.

## Methods
### Materials
Cobaltous nitrate hexahydrate (Co(NO₃)₂·6H₂O, AR), iron nitrate nonahydrate (Fe(NO₃)₃·9H₂O, AR), sodium phosphotungstate hydrate ($PW_{12}$-POM, Na₃PW₁₂O₄₀·xH₂O, AR), sodium phosphomolybdate hydrate ($PMo_{12}$-POM, Na₃PMo₁₂O₄₀·xH₂O, AR), sodium silicotungstate hydrate ($SiW_{12}$-POM, Na₄SiW₁₂O₄₀·xH₂O, AR), sodium hydroxide (NaOH, 95 wt.%), potassium hydroxide (KOH, 96 wt.%), ammonium fluoride (NH₄F, AR), urea (CO(NH₂)₂, AR), ruthenium oxide (RuO₂, AR), Pt/C (20 wt.%), nafion (5 wt.%), and sodium hypochlorite (NaClO, AR) were purchased from Aladdin Industrial Co. Hydrochloric acid (HCl, 98 wt.%) and anhydrous ethanol were sourced from Beijing Chemical Corp. NF (0.2 mm thick) was supplied by Qingyuan Metal Materials Co., Ltd. Seawater was collected from the Huangdao District of Qingdao, and ultrapure water (18.3 MΩ·cm) was used throughout.

### Synthesis of CoFe LDH/NF
CoFe LDH/NF was synthesized by hydrothermally treating a mixture of 3 mmol Co(NO₃)₂·6H₂O, 1 mmol Fe(NO₃)₃·9H₂O, and 5 mmol urea in 30 mL deionized water with a pre-cleaned 2 × 3 cm² NF at 120 °C for 10 h. The product was rinsed, dried at 60 °C, and shows a loading of ~1.65 mg cm⁻².

### Synthesis of POM-CoFe LDH/NF
POM-CoFe LDH/NF was obtained via 30 min immersion in 1 mM POM solution ($PW_{12}$-POM, $SiW_{12}$-POM, or $PMo_{12}$-POM), followed by drying at 60 °C. $PW_{12}$-CoFe LDH/NF shows a loading of ~1.94 mg cm⁻².

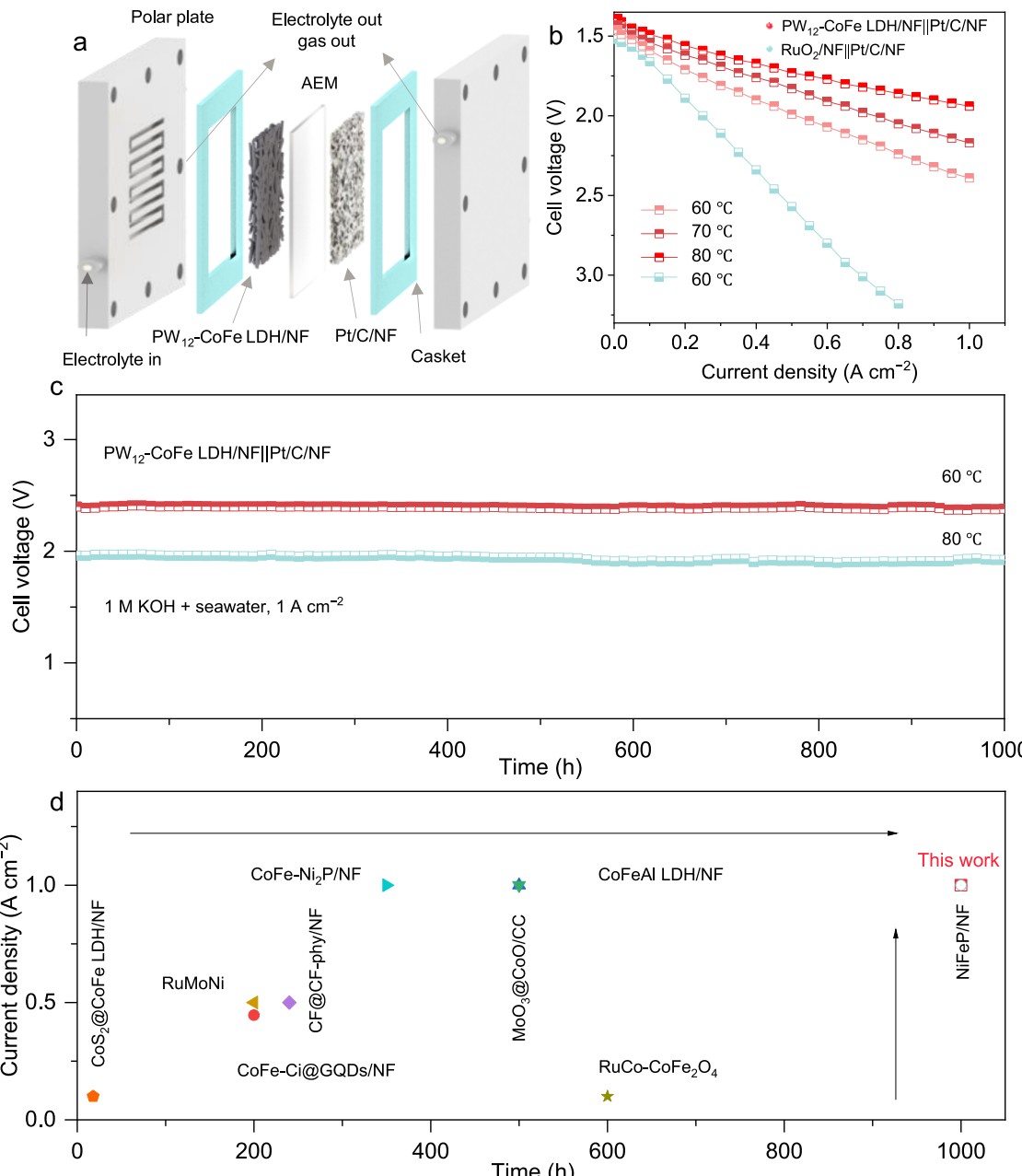

**Fig. 5 | Application of catalyst. a** Schematic of the flow electrolytic cell with symmetric seawater supply. **b** Polarization curves of PW₁₂-CoFe LDH/NF‖Pt/C/NF (60–80 °C) versus RuO₂/NF‖Pt/C/NF (60 °C) without *iR* compensation. **c** Continuous electrolysis tests at 1.0 A cm⁻² in 1 M KOH + seawater for PW₁₂-CoFe LDH/NF‖Pt/C/NF at 60 and 80 °C without *iR* compensation. **d** Comparison of the electrolysis durability of PW₁₂-CoFe LDH/NF anodes with that of recently reported ASO anodes evaluated in flow cells. Source data are provided as a Source Data file.

## Preparation of RuO₂/NF

RuO₂/NF electrode was prepared following our previously reported work[35], in which commercially available RuO₂ powders were uniformly deposited onto pre-cleaned NF with a mass loading identical to that of CoFe LDH/NF.

## Characterizations

X-ray diffraction (XRD, Bruker D8 Advance) was employed to characterize the crystal structures. Morphological and atomic-scale details were investigated by scanning electron microscopy (SEM, ZEISS Gemini SEM 300), transmission electron microscopy (TEM, JEOL JEM-F200), and aberration-corrected scanning transmission electron microscopy (AC-STEM, JEOL JEM-ARM200F). Surface chemical states were analyzed using X-ray photoelectron spectroscopy (XPS, Thermo

Scientific K-Alpha, Al Kα source). Ex situ X-ray absorption spectroscopy (XAS) tests at the Co and Fe K-edges and W L₃-edge were performed at beamline 1W1B of the Shanghai Synchrotron Radiation Facility (SSRF). ICP-OES (Agilent 5110) was used to quantify the time-dependent leaching of Co and Fe in alkaline seawater. UV–visible absorbance spectra were recorded on a Shimadzu UV-1800 spectro-photometer, and surface elemental analysis was conducted via time-of-flight secondary ion mass spectrometry (TOF-SIMS, PHI TRIFT V nanoTOF).

## Electrochemical measurements

Alkaline seawater oxidation experiments were conducted at ambient temperature using CHI 660E and CHI 760E electrochemical work-stations. Electrode stability was assessed under constant current

charging using a Land CT2001A system (Wuhan, China). A conventional three-electrode configuration was adopted, employing PW$_{12}$-CoFe LDH/NF, CoFe LDH/NF, RuO$_2$/NF, or NF as the working electrode, Hg/HgO as the reference electrode, and a graphite rod as the counter electrode. The electrolytes used included 1 M KOH, simulated seawater (1 M KOH + 0.5 M NaCl), and alkaline seawater (1 M KOH + seawater), with a volume of 50 mL. To remove excess Mg$^{2+}$ and Ca$^{2+}$, natural seawater was pretreated with an appropriate amount of Na$_2$CO$_3$. The 1 M KOH + seawater solution was prepared by dissolving 56.11 g of KOH in 1 L of pretreated seawater, followed by stirring and sonication. All electrolytes were freshly prepared and used within 24 h. The pH of the alkaline seawater electrolyte was measured to be 13.98 ± 0.01. Before conducting measurements, the electrodes underwent activation via cyclic voltammetry (CV) at a scan rate of 10 mV s$^{-1}$. All electrode potentials were recalculated against RHE using the conversion formula: E$_{RHE}$ = E$_{Hg/HgO}$ + 0.098 + 0.059 × pH. Calibration of the Hg/HgO electrode was performed in a H$_2$ atmosphere with a Pt wire serving as the working electrode. iR compensation was performed as E$_{corr}$ = E−iR, where E is the measured potential, R is the solution resistance, and i is the operating current. Impedance spectra were recorded between 10 kHz and 0.01 Hz with a 5 mV amplitude perturbation.

## Turnover frequency (TOF) calculation

The TOF was calculated using TOF = A·j/4Fm, where A is the geometric area of electrode, j is the current density, 4 corresponds to the number of electrons per mole of O$_2$, F is the Faraday constant (96,485 C mol$^{-1}$), and m represents the active site concentration. The value of m was determined from the slope of the oxidation peak current vs. scan rate, using Slope = n$^2$F$^2$m/4RT.

## Specific activity calculation

The specific activity was calculated by normalizing the j to the electrochemically active surface area (ECSA). The ECSA was estimated from the double-layer capacitance (C$_{dl}$), obtained from CV curves in the non-Faradaic region between 1.04 and 1.14 V vs. RHE. The capacitive current (Δj/2) at 1.09 V was plotted against scan rates to determine C$_{dl}$ as the slope. ECSA was then calculated using the equation ECSA = A·C$_{dl}$/C$_s$, where C$_s$ is 0.04 mF cm$^{-2}$.

## In situ Raman tests

In situ Raman spectra were acquired using a 532 nm laser (LabRAM HR Evolution, 50× objective) in a custom cell with CoFe LDH/NF or PW$_{12}$-CoFe LDH/NF as the working electrode, Pt counter, and Hg/HgO reference in 1 M KOH + seawater. Potential-dependent tests were run from OCP to 1.62 V vs. RHE on a CHI 660E system.

## In situ XAS tests

Co and Fe K-edge XAS spectra were collected at SSRF beamline 1W1B using a Si(111) monochromator and Co/Fe foils for energy calibration. Measurements were conducted under electrochemical control from OCP to 1.62 V vs. RHE using an electrochemical workstation. The catalyst-coated electrodes served as the working electrodes, while a Pt wire and an Hg/HgO electrode functioned as the counter and reference electrodes, respectively. Signal acquisition was carried out using standard ion chambers.

## Calculation methods

First-principles calculations were performed using the Vienna Ab initio Simulation Package (VASP) 6.3.2, employing spin-polarized density functional theory within the generalized gradient approximation and using the Perdew-Burke-Ernzerhof functional[56,57]. Core-electron interactions were described via projector augmented-wave pseudopotentials, and van der Waals interactions were included using the DFT-D3 method[58–60]. A 15 Å vacuum layer, 500 eV plane wave cutoff, and a 1 × 1 × 1 Monkhorst-Pack k-point grid were applied[61]. The self-

consistent and geometry optimization thresholds were set at 1 × 10$^{-5}$ eV and 0.03 eV·Å$^{-1}$, respectively. Adsorption energy (E$_{ads}$) was calculated as: E$_{ads}$ = E$_{total}$ − E$_{substrate}$ − E$_{adsorbate}$. Gibbs free energy (ΔG) was computed as: ΔG = ΔE + ΔZPE − TΔS, where ΔE is the DFT-calculated energy, ΔZPE is the zero-point energy correction, and TΔS accounts for entropy. It should be noted that the current computational models do not explicitly include solvation effects. Real electrochemical conditions involving solvated OH$^-$ and Cl$^-$ may substantially differ due to solvent effects, such as desolvation and dielectric-induced electron density variations. Therefore, direct quantitative comparisons between computed results and experimental data should be made cautiously. To facilitate structural visualization, the optimized CONTCAR for PW$_{12}$-CoFe LDH is included in Supplementary Data 1.

## Fabrication of MEA

For MEA construction, PW$_{12}$-CoFe LDH/NF and Pt/C/NF (geometric area: 1 cm$^2$) were employed as the anode and cathode, respectively. These electrodes were separated by an anion exchange membrane (60 μm, 1.2 × 1.2 cm$^2$) that had been pretreated in 1 M KOH for over 24 h and thoroughly rinsed. The assembled cell was operated under a continuous flow of alkaline seawater (50 mL min$^{-1}$), and its performance was tested across 60–80 °C using a GW Instek PSW 80-13.5.

## Data availability

The source data generated in this study are provided in the Source Data file. Source data are provided in this paper.

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

## Acknowledgements

X.S. acknowledges the funding support from the Free Exploration Project of Frontier Technology for Laoshan Laboratory (No. 16-02). F.L. acknowledges the funding support from the Science and Technology Program of Tibet (No. XZ202201ZY0002G). Q.Y. acknowledges the funding support from the Natural Science Foundation of China (No. 22473071) and the NSFC Center for Single-Atom Catalysis (No. 22388102).

## Author contributions

X.H. and X.S. designed this research. X.H. and X.S. wrote the manuscript. X.H., Y.Y. and M.Z. performed material synthesis, characterizations, and performance tests. X.H., Y.Y. and L.Z. conceived and completed all the schematic drawings. Y.Z. and Q.Y. conducted theoretical calculations. Y.R., K.D., H.T., J.N., X.Z., H.L. and B.Y. participated in discussions. F.L., B.T. and X.S. supervised the research. All authors contributed and reviewed the manuscript.

## Competing interests

The authors declare no competing interests.
