## [Transparent Peer Review file · Nature Communications]

Engineered PW12-polyoxometalate docked Fe sites on CoFe hydroxide anode for durable seawater electrolysis

Corresponding Author: Professor Xuping Sun

Version 0:

Reviewer comments:

Reviewer #1

(Remarks to the Author)

This manuscript reports the preparation of POM-CoFe LDH electrocatalyst supported on Ni foam. Its catalytic performance for OER in alkaline seawater is evaluated and a stable response under large current density is demonstrated. However, there are some questions in this manuscript, which makes it premature for publication in Nature Communications at this stage.

1. The authors suggest POM-docked CoFe LDH to be a good architecture for seawater oxidation. However, there are many kinds of POMs. Why choosing Na₃PW₁₂O₄₀? How would the structure and the catalytic performance of POM-CoFe LDH be if using other POMs? Systematical study should be carried out.
2. The authors declare an interaction between Fe and W in POM-CoFe LDH. However, they do not provide enough evidence. Only the states of Fe are characterized. The XAS and XPS of W should also be characterized and analyzed.
3. The authors think the average oxidation state of Fe in POM-CoFe LDH is 3.51. But Fe(IV) is quite unstable and can only exist under harsh oxidation condition. How can Fe preserve such high valence without OER test?
4. Since the authors use the valence changes of Co and Fe during OER as a key evidence to explain the catalytic mechanism, they should be characterized by in-situ XAS rather than ex-situ XPS.
5. Ni foam is used as the substrate in this manuscript. But Ni is also an OER active species. NiFe (oxy)hydroxides are well known high performance OER catalysts. The authors should use other supports to eliminate the influence of Ni, otherwise the effect of Ni should be discussed.
6. Why does the POM interact with Fe rather than Co? In-depth mechanism discussion should be provided.
7. Why choosing Co to be the OER active sites for DFT calculation? What are the differences between Co sites and Fe sites for the adsorption/desorption of OER intermediates?
8. The authors suggest that "For POM-CoFe LDH/NF, this structural transformation occurs at more positive potentials, suggesting that POM accelerates dehydrogenation and promotes the formation of catalytically favorable metal oxyhydroxides under ASO conditions (Fig. 3c, d)". How can the more positive transformation potentials indicate the favorable formation of metal oxyhydroxides?
9. The authors declare that " ΔG^*OH decreases considerably from 1.087 eV to -0.836 eV with the POM introduction, whereas ΔG^*Cl only decreases to -0.226 eV. This stark contrast (-0.836 eV vs. -0.226 eV) underscores a pronounced selectivity for *OH adsorption over *Cl on CoFeOOH surfaces". However, the significantly decreased free energy for the adsorption of *OH would generally indicate higher energy barrier for OER when considering the conversion of OER intermediates. Would the rate-determining step for OER still be faster than Cl oxidation under such condition?
10. MEA should be evaluated under 60-80 °C rather than room temperature.

Reviewer #2

(Remarks to the Author)

The authors described the polyoxometalate (POM)-docked CoFe-LDH electrocatalyst for efficient and robust alkaline seawater oxidation. The POM modification strategy not only effectively modulates the electronic structure of the catalyst to enhance its activity, but also extends the catalyst's lifespan by repelling chloride ions. The ultralow overpotential (368 mV @ 1000 mA cm⁻²) and superior operation stability (1300 h @ 1 A cm⁻² and 600 h @ 2 A cm⁻²) are impressive. However, there are some problems need to be addressed before considering this manuscript for publication. I recommend that this manuscript can be accepted for publication after a major revision. The details of my comments are listed below.

1. The experimental methods and physical characterization procedures should be explicitly provided. Additionally, the data analysis protocols and procedures need to be detailed, such as the double layer capacitance and TOF calculation, which should be clearly stated in the method section. It was particularly noted that the double layer capacitance was plotted as Δj rather than $\Delta j/2$ (Supplementary Fig. 18).
2. Further clarification on the iR correction procedure is required, with a clear indication of whether the data has been compensated, and meanwhile avoiding overcompensation of iR potential drop.
3. More rigorous and detailed analyses need to be provided to reach the final conclusion. For example:
 - a) Why are reduced phase angle observed in Fig. 2B due to surface charge accumulation rather than the acceleration of electrons transfer?
 - b) The manuscript emphasizes the positive shift of oxidation peaks (Fig. 3c, d), which may contradict with the active phase of Co(Fe)OOH for efficient OER. The authors should explain it. Meanwhile, why such a delay of CoFe-LDH oxidation accelerates deprotonation process, need careful discussion.
 - c) Why did the authors conclude that POM modification increases the acidity of Co sites when there was no apparent change in the valence state and coordination structure? And why a significant increase of Fe oxidation state does not result in a more acidic site?
 - d) Page6 line123, "a slight shift" and "substantial electronic interplay and charge transfer" seem to be paradoxical. Co sites are inferred as the active sites, as Fe is largely blocked by POM. The authors should explain why negligible changes of Co would drive efficient OER. During OER the oxidation state of Co would be increased as expected, how can we correlate this oxidation-state change to acidity?
 - e) The question remains as to why the docking of POM on Fe atoms can also prevent Co ions from losing, which should be discussed more carefully.
 - f) No jagged edges can be clearly distinguished in SEM and AC-STEM images (Page6 line111), confirm it.
 - g) Fig. S6, I cannot identify significantly enhanced W=O asymmetric stretching.
4. The first two spectra of TOF-SIMS lack clear identification of ion types, as they are presented as intensity plots. It is essential to provide more specific labeling and rigorous interpretation of the results.
5. There is a lack of characterization information on the samples after long-term testing to demonstrate the structural stability of the catalyst. If possible, XRD, TEM, and XPS data should be provided.
6. It is strange that, while extensive evidence is presented for the chemical valence and coordination environment of Co and Fe, there is a lack of explanation concerning P and W elements. During operation, especially at ampere-level current densities, would POM decompose? Would the Keggin structure keep stable in alkaline as this unique structure is generally stable in acidic environment? Is the real protective species such as PO₄³⁻ and WO₄²⁻ formed after POM degradation? Therefore, the stability or evolution of POM needs to be addressed, particularly as phosphate has been proven to effectively repel Cl⁻. If possible, control experiments using PO₄³⁻ and WO₄²⁻ should be added.
7. The citation in Fig. 2e is not comprehensive, the authors should update it, for example 1.25 A cm⁻² for 2800 hours in Ref. 50. The current density of Ref. 50 in Fig. 5c should also be double checked.
8. The potential configuration of POM on CoFe-LDH should be carefully discussed, with only one coordination site as the authors proposed in Fig. S27, or more sites are possible. The unique roles of POM in the reaction mechanism, electronic modulation, protection should be intensified.

Reviewer #3

(Remarks to the Author)

The authors of 'Engineered polyoxometalate-docked Fe sites on CoFe hydroxide anodes towards industrial-level alkaline seawater electrolysis' have used a variety of different techniques to prove the effect of polyoxometalates on repelling Cl⁻ adsorption and increase at the same time faradaic efficiency towards OER and stability. However, I have a few key questions.

- 1) In Fig. 1j the authors calculate the Fe oxidation state based on the peak of the 1st derivative which obviously is not well smoothed and the authors chose the far left peak for the CoFe-LDH catalyst and far right for the POM-CoFe-LDH, which meant a E₀ of 7124 and 7128 respectively giving huge difference in the expected oxidation state, from 2.62 to 3.51. One can see that with a correct smoothing factor the oxidation states are very similar.
- 2) In Fig. 1k the authors suggest a Fe-O-Co/W bond at around 2.5Å, however the same peak can be seen for the FeO and the Fe₂O₃. Why is that?
- 3) Additionally, the calculations on ΔG_{OH} for the 2 catalysts also give abnormally different energy values with and without POM. Also OH adsorption by itself does not solely govern catalytic activity.

In general this is a very solid work that after a few modifications looks like it can have a significant impact on sea water electrolysis by easy synthesis of catalysts and a simple surface modification.

Reviewer #4

(Remarks to the Author)

The authors have developed poly-oxometalate (POM) based POM-CoFe LDH/NF anode for possible use in electrolyzers that use saline water with high Cl⁻ content. The POM coating provides stability to the anode by avoiding Cl⁻ mediated metal leaching and enhances dehydrogenation by modulating electron density on active Co-sites. POM-based materials for various electrolyzers have been developed over the years, and I think the authors should have a look at the latest review article, <https://doi.org/10.1039/D3QM01000G>.

In any case, the work presented here is well executed and detailed studies on each aspect of the electrochemical processes

have been carried out. I do not have many comments except a minor query related to the computational part of the work -

All the DFT calculations were carried out in the gas phase at 0 K temperature. Although temperature correction takes care of the ΔS part, how does the model describe solution phase reactions on the electrode surface? Typically the OH⁻ and Cl⁻ ions are well-solvated in a realistic system, and their attachment will be largely affected by a desolvation-type attachment mechanism. The solvent dielectric might impact the charge density distribution significantly. At least, the implicit model description is necessary to incorporate the solvent effect. A cautionary comment on this will be required while comparing experimental and computational outcomes.

Version 1:

Reviewer comments:

Reviewer #1

(Remarks to the Author)

Most of my concerns are addressed in the revised manuscript. But there are still some problems.

1. The authors synthesized and characterized CoFe LDH catalysts modified with PW₁₂-POM, SiW₁₂-POM, and PMo₁₂-POM. They declare that “unlike PW₁₂-POM, the incorporation of SiW₁₂-POM or PMo₁₂-POM shifts the Co 2p peaks to lower binding energies, suggesting electron accumulation at Co sites and a reduced oxidation state”. However, the shift of binding energy for Co is unobvious. Moreover, the binding energies of W and Mo for SiW₁₂-POM, and PMo₁₂-POM both shift negatively. How can the Co sites accumulate electrons under such condition? Besides, why does PW₁₂-POM show different performance compared with SiW₁₂-POM and PMo₁₂-POM? How would Si affect the interaction between different metal sites? What is the difference between W and Mo? More in-depth theoretical discussion should be provided to improve the scientific significance of this work.

2. In Figure 5, the operation voltage of the cell at 60 °C is higher than that of 25 °C in the previous version. The author should check the data and provide explanation for the strange performance. Moreover, some recent progresses should be added in Figure 5d, such as Nature, 2025, 639, 360.

Therefore, revision is still needed for this manuscript.

Reviewer #2

(Remarks to the Author)

The authors have addressed all of my concerns with the original manuscript. The revised manuscript is ready for publication.

Reviewer #3

(Remarks to the Author)

The authors have thoroughly answered all remarks and questions and as a result I believe this is a nice study on electrocatalysts for the sea water electrolysis .

Reviewer #4

(Remarks to the Author)

The authors have addressed my comments. I do not have any further comments on the manuscript, and I recommend it for publication in Nature Communication.

Version 2:

Reviewer comments:

Reviewer #1

(Remarks to the Author)

This manuscript can be accepted without further revision.

Point-by-Point Responses to Reviewers' Comments

We express our sincere gratitude to the editor and all reviewers for their invaluable feedback, which we have utilized to enhance the quality of our manuscript (NCOMMS-24-69947). The reviewer comments are presented in *italic* and **bold** font, while our responses, including the incorporation of additional figures, tables, descriptions, and other elements, are highlighted in **blue** text.

Point-by-point response to the reviewers #1

Reviewer #1 (Remarks to the Author): This manuscript reports the preparation of POM-CoFe LDH electrocatalyst supported on Ni foam. Its catalytic performance for OER in alkaline seawater is evaluated and a stable response under large current density is demonstrated. However, there are some questions in this manuscript, with makes it premature for publication in Nature Communications at this stage.

Response: We are grateful for the valuable comments and suggestions, which have led to substantial improvements in the manuscript. In response, we have conducted additional characterizations, refined the mechanistic discussion, and carefully revised the text for greater clarity and rigor. We believe these modifications strengthen the scientific impact and completeness of the study.

Comment 1: The authors suggest POM-docked CoFe LDH to be a good architecture for seawater oxidation. However, there are many kinds of POMs. Why choosing $\text{Na}_3\text{PW}_{12}\text{O}_{40}$? How would the structure and the catalytic performance of POM-CoFe LDH be if using other POMs? Systematical study should be carried out.

Response: We appreciate your insightful comments. $\text{Na}_3\text{PW}_{12}\text{O}_{40}$ (PW_{12}) was selected owing to its high negative charge density and proven redox flexibility (*Mater. Chem. Front.* **8** (2024) 732; *Chem. Soc. Rev.* **41** (2012) 7605; *Angew. Chem. Int. Edit.* **59** (2020) 20779), which we hypothesized would both repel chloride ions during alkaline seawater oxidation (ASO). To assess the effect of POM type, we synthesized CoFe LDH catalysts modified with PW_{12} -POM, SiW_{12} -POM, and PMo_{12} -POM under the same conditions. As shown in **Fig. R1** (Supplementary Fig. 21),

Fig. R1 Comparison of polarization curves for CoFe LDH/NF, SiW₁₂-CoFe LDH/NF, PMo₁₂-CoFe LDH/NF, and PW₁₂-CoFe LDH/NF with 100% *iR* compensation.

Fig. R2 Comparison of XPS spectra for SiW₁₂-CoFe LDH/NF and CoFe LDH/NF in the (a) Co 2*p* and (b) Fe 2*p* regions. (c) Comparison of XPS spectra for SiW₁₂-CoFe LDH/NF and SiW₁₂-POM in the W 4*f* region. Comparison of XPS spectra for PMo₁₂-CoFe LDH/NF and CoFe LDH/NF in the (d) Co 2*p* and (e) Fe 2*p* regions. (f) Comparison of XPS spectra for PMo₁₂-CoFe LDH/NF and PMo₁₂-POM in the Mo 3*d* region.

only the PW₁₂-CoFe LDH/NF showed improved ASO activity. Both SiW₁₂-POM and PMo₁₂-POM-modified catalysts exhibited diminished performance (Fig. R2; Supplementary Fig. 22). XPS analysis indicates that, unlike PW₁₂-POM, the incorporation of SiW₁₂-POM or PMo₁₂-POM shifts the Co 2p peaks to lower binding energies, suggesting electron accumulation at Co sites and a reduced oxidation state. This change likely impairs catalytic efficiency. These comparative results have been included in the revised manuscript to support the choice of PW₁₂-POM.

Comment 2: The authors declare an interaction between Fe and W in POM-CoFe LDH. However, they do not provide enough evidence. Only the states of Fe are characterized. The XAS and XPS of W should also be characterized and analyzed.

Fig. R3 (a) Normalized W L₃-edge XANES spectra of PW₁₂-CoFe LDH, CoFe LDH, WO₃, and W foil. (b) FT-EXAFS spectra of PW₁₂-CoFe LDH, PW₁₂-POM, WO₃, and W foil. (c) XPS spectra for PW₁₂-CoFe LDH/NF (top) and PW₁₂-POM (bottom) in the W 4f regions.

Response: We appreciate your insightful comments. We have now performed detailed XAS and XPS characterizations of W to provide further evidence of the Fe and W interaction. The W L₃-edge XANES spectra show a clear shift to lower energy relative to PW₁₂-POM (**Fig. R3a**; Supplementary Fig. 8), which is consistent with the W 4f XPS signal (**Fig. R3c**; Supplementary Fig. 5c). Moreover, EXAFS analysis reveals a pronounced increase in the W–O coordination number and a significant elongation of both W–O and W–O–W bonds (**Fig. R3b**; Supplementary Fig. 12), indicating strong interfacial coordination. These findings, together with the oxidation state profiles of Co and Fe and the DFT-predicted charge redistribution, confirm that PW₁₂-POM coordinates with Fe through its terminal W=O groups. All supporting data have been added to the revised manuscript.

Comment 3: The authors think the average oxidation state of Fe in POM-CoFe LDH is 3.51. But Fe(IV) is quite unstable and can only exist under harsh oxidation condition. How can Fe preserve such high valence without OER test?

Response: We appreciate your insightful comments. We consider that the stabilization of Fe in a high oxidation state arises from the strong electron-accepting nature of PW₁₂-POM, which facilitates charge transfer and electronic delocalization. Our DFT results show that coordination between PW₁₂-POM and Fe leads to substantial electron transfer, and this interaction is energetically favorable enough to maintain Fe in a high-valent state.

Comment 4: Since the authors use the valence changes of Co and Fe during OER as a key evidence to explain the catalytic mechanism, they should be characterized by in-situ XAS rather than ex-situ XPS.

Response: We appreciate your insightful comments. As suggested, we have conducted in situ XAS measurements to track the valence evolution of Co and Fe during ASO (**Fig. R4**). The Co K-edge of PW₁₂-CoFe LDH shows a more prominent and rapid positive shift with increasing potential compared to CoFe LDH, indicating enhanced redox activity at Co sites (**Fig. 3e**; Supplementary Fig. 32a, b). In contrast, the Fe K-edge remains largely

unchanged in $\text{PW}_{12}\text{-CoFe LDH}$, while shifting gradually in CoFe LDH , confirming that $\text{PW}_{12}\text{-POM}$ effectively stabilizes Fe against oxidation during ASO (Supplementary Fig. 32c, d).

Fig. R4. In-situ Co K-edge XANES spectra of (a) CoFe LDH and (b) $\text{PW}_{12}\text{-CoFe LDH}$. In-situ Fe K-edge XANES spectra of (c) CoFe LDH and (d) $\text{PW}_{12}\text{-CoFe LDH}$.

Comment 5: Ni foam is used as the substrate in this manuscript. But Ni is also an OER active species. NiFe (oxy)hydroxides are well known high performance OER catalysts. The authors should use other supports to eliminate the influence of Ni, otherwise the effect of Ni should be discussed.

Response: We appreciate your insightful comments. While Ni foam can contribute to OER via NiFe (oxy)hydroxide formation, our data show negligible interference from the Ni substrate. SEM image reveals that without $\text{PW}_{12}\text{-POM}$, Ni foam undergoes visible corrosion within 80 hours, while with $\text{PW}_{12}\text{-POM}$, the framework is preserved after 1300 hours (Supplementary Fig. 34). This is supported by stable Ni peaks in XRD

(Supplementary Fig. 35a). Moreover, XAS, XPS, and ICP-OES analyses show strong coordination between PW_{12} -POM and Fe, which suppresses Fe leaching and inhibits the formation of NiFe (oxy)hydroxides. Therefore, the catalytic behavior we report is primarily associated with the POM-CoFe LDH itself, rather than contributions from Ni-derived species. Additionally, Ni foam is widely employed as a support in seawater oxidation studies, where anionic species have been shown to mitigate Cl^- -induced corrosion (*Nat. Sustain.* **7**, (2024) 158; *Nat. Commun.* **15**, (2024) 4712; *Nat. Commun.* **15**, 4712 (2024); *ACS Nano* **19**, (2025) 1530; *Small* **18**, (2022) 2203852). We believe its use here is justified and does not interfere with the interpretation of catalytic results.

Comment 6: Why does the POM interact with Fe rather than Co? In-depth mechanism discussion should be provided.

Fig. R5. (a) XPS spectrum of PW_{12} -Co(OH)₂ in the W 4f region. (b) Structural diagrams of PW_{12} -CoFe LDH connected via Fe and Co sites.

Response: We appreciate your insightful comment. A detailed discussion has been added to the revised manuscript. Our XPS analysis reveals that the introduction of PW_{12} -POM leads to only slight shifts in the Co 2p peaks, whereas the Fe 2p peaks shift markedly toward higher binding energies. These changes are accompanied by

corresponding shifts to lower binding energies in the W 4*f* and P 2*p* regions, indicating electron transfer from Fe to PW₁₂-POM. XANES results further support this observation. Additionally, PW₁₂-Co(OH)₂ shows negligible W 4*f* signal intensity (Fig. R5a; Supplementary Fig. 6), suggesting minimal interaction between PW₁₂-POM and Co. DFT calculations also confirm this selectivity, showing that the binding energy between PW₁₂-POM and Fe (0.103 eV) is substantially lower than that with Co (0.864 eV) (Fig. R5b; Supplementary Fig. 29). These combined results indicate that PW₁₂-POM interacts with Fe rather than Co.

Comment 7: Why choosing Co to be the OER active sites for DFT calculation? What are the differences between Co sites and Fe sites for the adsorption/desorption of OER intermediates?

Fig. R6. Gibbs free-energy profile for the four-step OER pathway on the Fe and Co sites for PW₁₂-CoFeOOH.

Response: We appreciate your thoughtful comments. Based on systematic ex situ and in situ results, we found that PW₁₂-POM coordinates selectively with Fe, leading to its electronic stabilization. Consequently, Co is identified as the likely active site for OER. In our study, we explicitly investigated the effect of PW₁₂-POM on Co sites and their

interaction with OER intermediates. To compare with Fe, we also performed DFT calculations considering Fe as the active center. As shown in Fig. R6, while Fe sites exhibit stronger OH⁻ adsorption, they suffer from an exceptionally high energy barrier in the *OOH to O₂ conversion step, which is the rate-limiting step of the reaction. This makes O₂ desorption highly unfavorable. According to the Sabatier principle, such overly strong binding reduces catalytic efficiency. In contrast, Co provides a more balanced interaction with intermediates, thus enabling more favorable OER kinetics.

Comment 8: *The authors suggest that “For POM-CoFe LDH/NF, this structural transformation occurs at more positive potentials, suggesting that POM accelerates dehydrogenation and promotes the formation of catalytically favorable metal oxyhydroxides under ASO conditions (Fig. 3c, d)”. How can the more positive transformation potentials indicate the favorable formation of metal oxyhydroxides?*

Response: We are grateful for your comment and sincerely apologize for the oversight. Upon careful re-examination, we confirm that the structural transformation in POM-CoFe LDH/NF occurs at lower potentials. We have corrected this statement and revised the manuscript to accurately reflect the experimental observations.

Comment 9: *The authors declare that “ ΔG^*OH decreases considerably from 1.087 eV to -0.836 eV with the POM introduction, whereas ΔG^*Cl only decreases to -0.226 eV. This stark contrast (-0.836 eV vs. -0.226 eV) underscores a pronounced selectivity for *OH adsorption over *Cl on CoFeOOH surfaces”. However, the significantly decreased free energy for the adsorption of *OH would generally indicate higher energy barrier for OER when considering the conversion of OER intermediates. Would the rate-determining step for OER still be faster than Cl oxidation under such condition?*

Response: We thank you for raising this insightful point. While the adsorption free energy for *OH decreases notably with POM introduction (-0.836 eV), our additional calculations (Fig. R7a; Supplementary Fig. 39) show that this does not result in a significantly increased energy barrier for the conversion of OER intermediates.

Specifically, the rate-determining *O to *OOH step has a barrier of 2.425 eV for PW₁₂-CoFeOOH, which is lower than that of CoOOH (2.487 eV). Moreover, this value remains well below the barrier for Cl⁻ oxidation (*Cl + Cl⁻ to Cl₂, 2.946 eV) (Fig. R7b; Supplementary Fig. 40), confirming that OER remains the thermodynamic favored pathway.

Fig. R7. (a) Gibbs free-energy profile for the four-step OER pathway on the Co sites of PW₁₂-CoFeOOH versus CoFeOOH. (b) Gibbs free-energy profile for the two-step ClOR pathway on the Co sites of PW₁₂-CoFeOOH versus CoFeOOH.

Comment 10: MEA should be evaluated under 60-80 °C rather than room temperature.

Response: We appreciate your valuable suggestion. As suggested, we have re-evaluated the MEA performance of PW₁₂-CoFe LDH/NF||Pt/C/NF at 60 to 80 °C. The new data, presented in Fig. R8 (Fig. 5), confirm that it maintains stable and efficient

operation under high temperatures. These results have been incorporated into the revised manuscript.

Fig. R8. (a) Schematic of the flow electrolytic cell with symmetric seawater supply. (b) Polarization curves of $PW_{12}\text{-CoFe LDH/NF} || \text{Pt/C/NF}$ (60–80 °C) versus $\text{RuO}_2/\text{NF} || \text{Pt/C/NF}$ (60 °C) without iR compensation. (c) Continuous electrolysis tests at 1.0 A cm^{-2} in 1 M KOH + seawater for $PW_{12}\text{-CoFe LDH/NF} || \text{Pt/C/NF}$ at 60 and 80 °C. (d) Comparison of the electrolysis durability of $PW_{12}\text{-CoFe LDH/NF}$ anodes with that of recently reported ASO anodes evaluated in flow cells.

Point-by-point response to the reviewers #2

Reviewer #2 (Remarks to the Author): *The authors described the polyoxometalate (POM)-docked CoFe-LDH electrocatalyst for efficient and robust alkaline seawater oxidation. The POM modification strategy not only effectively modulates the electronic structure of the catalyst to enhance its activity, but also extends the catalyst's lifespan by repelling chloride ions. The ultralow overpotential (368 mV @ 1000 mA cm⁻²) and superior operation stability (1300 h @ 1 A cm⁻² and 600 h @ 2 A cm⁻²) are impressive. However, there are some problems need to be addressed before considering this manuscript for publication. I recommend that this manuscript can be accepted for publication after a major revision. The details of my comments are listed below.*

Response: Thank you for the detailed assessment and valuable suggestions. We have carefully addressed all comments through a comprehensive revision of the manuscript. New data, mechanistic insights, and methodological clarifications have been added where appropriate. We believe these revisions enhance both the robustness and clarity of our study.

Comment 1: *The experimental methods and physical characterization procedures should be explicitly provided. Additionally, the data analysis protocols and procedures need to be detailed, such as the double layer capacitance and TOF calculation, which should be clearly stated in the method section. It was particularly noted that the double layer capacitance was plotted as Δj rather than $\Delta j/2$ (Supplementary Fig. 18).*

Response: Thank you for this valuable suggestion and for pointing out the issue with the C_{dl} plot. In the revised manuscript, we have provided comprehensive descriptions of the experimental methods and physical characterization techniques. Additionally, we have clarified the data analysis protocols, including the calculations of turnover frequency (TOF) and double-layer capacitance (C_{dl}), which are now explicitly stated in the **Methods** section. The revised text is as follows:

“Turnover frequency (TOF) calculation. The TOF was calculated using $TOF = A_j/4Fm$,

where A is the geometric area of electrode, j is the current density, 4 corresponds to the number of electrons per mole of O_2 , F is the Faraday constant ($96,485 \text{ C mol}^{-1}$), and m represents the active site concentration (mol). The value of m was determined from the slope of the oxidation peak current vs. scan rate, using $\text{Slope} = n^2F^2m/4RT$.”

“**Specific activity calculation.** The specific activity was determined by normalizing the j to electrochemically active surface area (ECSA). The ECSA was calculated based on the C_{dl} , measured through CV curves from 1.04 to 1.14 V vs. RHE, where no Faradaic current occurs. C_{dl} was plotted as $\Delta j/2$ at 1.09 V vs. RHE against scan rates to obtain the slope. ECSA was then derived using the formula $\text{ECSA} = AC_{dl}/C_s$, with C_s being 0.04 mF cm^{-2} .”

The previously noted error in plotting C_{dl} as Δj has been corrected to $\Delta j/2$ in **Fig. R9** (Supplementary Fig. 23).

Fig. R9. The electrochemical double-layer capacitance (C_{dl}) values for (a) PW₁₂-CoFe LDH/NF and (b) CoFe LDH/NF electrodes.

Comment 2: *Further clarification on the iR correction procedure is required, with a clear indication of whether the data has been compensated, and meanwhile avoiding overcompensation of iR potential drop.*

Response: Thank you for pointing this out. We have now included a clear explanation of the iR compensation procedure in the revised manuscript. The applied correction ensures accuracy without overcompensating the potential drop, as shown in Fig. 2a and Supplementary Fig. 21.

Comment 3: More rigorous and detailed analyses need to be provided to reach the final conclusion. For example:

a) Why are reduced phase angle observed in Fig. 2b due to surface charge accumulation rather than the acceleration of electrons transfer?

Response: Thank you for raising this point. To clarify, we have revised the manuscript as follows:

“In situ Bode phase plots further indicate that the transition peaks in the 10^{-1} – 10^1 Hz frequency range differ between these electrodes. The smaller phase angle of PW₁₂-CoFe LDH implies more charges are engaged in Faradaic processes at the interface. Notably, the phase angle of PW₁₂-CoFe LDH/NF decreases most rapidly, suggesting easier polarization and faster reaction kinetics (Fig. 2b).”

b) The manuscript emphasizes the positive shift of oxidation peaks (Fig. 3c, d), which may contradict with the active phase of Co(Fe)OOH for efficient OER. The authors should explain it. Meanwhile, why such a delay of CoFe-LDH oxidation accelerates deprotonation process, need careful discussion.

Response: We are grateful for your comment and sincerely apologize for the oversight. Upon careful re-examination, we confirm that the structural transformation in POM-CoFe LDH/NF occurs at lower potentials. We have corrected this statement and revised the manuscript to accurately reflect the experimental observations.

c) Why did the authors conclude that POM modification increases the acidity of Co sites when there was no apparent change in the valence state and coordination structure? And why a significant increase of Fe oxidation state does not result in a more acidic site?

Response: Thank you for your thoughtful comment. While the oxidation state and coordination environment of Co appear largely unchanged in the static measurements, in situ XAS reveals that POM coordination at Fe sites accelerates Co oxidation during ASO (Fig. R4, Supplementary Fig. 32), which promotes deprotonation and enhances the acidity of Co centers (Fig. 3e). Differential charge density analysis further confirms

electron withdrawal from Co via the Fe-bound POM unit (Supplementary Fig. 41). DFT calculations also demonstrate stronger *OH adsorption on Co after POM incorporation (Fig. 4d,e), supporting the increased acidity. Although Fe exhibits a higher oxidation state, it remains stable during ASO and binds *OH too strongly (Fig. R4d), leading to an ultrahigh energy barrier (4.122 eV) for the *OOH to O₂ step (Fig. R6), which hinders catalytic turnover. According to the Sabatier principle, such overbinding suppresses OER efficiency. Thus, it is the modulation of Co electronic structure by Fe-bound POM, that accounts for the increased catalytic acidity and activity.

d) Page6 line123, “a slight shift” and “substantial electronic interplay and charge transfer” seem to be paradoxical. Co sites are inferred as the active sites, as Fe is largely blocked by POM. The authors should explain why negligible changes of Co would drive efficient OER. During OER the oxidation state of Co would be increased as expected, how can we correlate this oxidation-state change to acidity?

Response: Thank you for your thoughtful comment. As elaborated in Response 3c, PW₁₂-POM coordination selectively stabilizes Fe, thereby directing the oxidative response toward Co during ASO. Although the shift in Co 2p binding energy is modest, in situ XAS confirms a pronounced increase in Co oxidation state in PW₁₂-CoFe LDH compared to the pristine sample. This oxidation facilitates deprotonation and enhances the Lewis acidity of Co, promoting *OH over *Cl adsorption. Thus, even subtle spectral changes reflect substantial electronic modulation that translates into improved OER activity. We have revised the manuscript to clarify this interpretation.

e) The question remains as to why the docking of POM on Fe atoms can also prevent Co ions from losing, which should be discussed more carefully.

Response: Thank you for this important point. As discussed in Responses 3c and 3d, PW₁₂-POM coordination stabilizes the Fe oxidation state and modulates the electronic environment of adjacent Co sites. To further understand this effect, we conducted DFT calculations comparing CoFeOOH and PW₁₂-CoFeOOH. The results show that upon POM incorporation, the free energy for *OH adsorption at Co sites decreases

significantly from 1.087 eV to -0.836 eV, while the change in *Cl adsorption energy is modest (from -0.083 eV to -0.226 eV) (Fig. 4d,e; Supplementary Fig. 38). This substantial difference in adsorption energetics confirms a strong selectivity for *OH over *Cl , which helps suppress chloride-induced Co leaching and contributes to the improved structural stability.

f) No jagged edges can be clearly distinguished in SEM and AC-STEM images (Page 6 line 111), confirm it.

Response: We appreciate your careful observation. Upon re-examination of the SEM and AC-STEM images, we agree that jagged edges are not clearly distinguishable. The manuscript has been corrected accordingly to avoid any misinterpretation.

g) Fig. S6, I cannot identify significantly enhanced W=O asymmetric stretching.

Response: Thank you for your careful observation. Upon careful reassessment of Fig. S6, we acknowledge that the W=O asymmetric stretching signal is not strongly enhanced on its own. Instead, the key feature lies in the relative increase in intensity of the asymmetric stretching (AS(W=O)) compared to the symmetric stretching (SS(W=O)) for PW_{12} -CoFe LDH. This distinction has been clarified in the revised text.

Comment 4: The first two spectra of TOF-SIMS lack clear identification of ion types, as they are presented as intensity plots. It is essential to provide more specific labeling and rigorous interpretation of the results.

Response: We appreciate your helpful suggestion. We have revised the manuscript to clearly state that the first two TOF-SIMS spectra are total ion spectra. Specific ion types have now been labeled accordingly, and a more comprehensive interpretation of the spectral features has been included to ensure accurate representation and analysis.

Comment 5: There is a lack of characterization information on the samples after long-term testing to demonstrate the structural stability of the catalyst. If possible, XRD, TEM, and XPS data should be provided.

Response: We appreciate your insightful comments. As suggested, we have now

provided comprehensive post-electrolysis characterizations, including XRD, TEM, Raman, and XPS analyses, which are discussed in detail in the revised manuscript and Supplementary Information.

Fig. R10. (a) XRD pattern of PW₁₂-CoFe LDH/NF and (b) TEM image of PW₁₂-CoFe LDH after 1300 hours of electrolysis.

Fig. R11. (a) Raman spectrum of PW₁₂-CoFe LDH/NF after 1300 hours of electrolysis. Comparison of XPS spectra for PW₁₂-CoFe LDH/NF (top, after reaction; bottom, before reaction) in the (b) Fe 2p, (c) Co 2p, and (d) W 4f regions after 1300 hours of electrolysis.

“XRD pattern and TEM image from long-term electrolysis confirm that PW₁₂-CoFe LDH retains its layered structure (Fig. R10) (Supplementary Fig. 35). Raman spectrum

indicates that PW_{12} -POM remains stably present and that most of the CoFe LDH is converted into metal oxyhydroxides (**Fig. R11a**) (Supplementary Fig. 36a). XPS analysis further shows that PW_{12} -POM stabilizes Fe and confirms extensive formation of metal oxyhydroxides ((**Fig. R11b-d**; Supplementary Fig. 36b–d)."

"After 1300 hours of electrolysis, the XRD pattern of PW_{12} -CoFe LDH/NF shows that the (003) and (006) planes are retained while other diffraction peaks disappear (Supplementary Fig. 35a), indicating that the layered structure of PW_{12} -CoFe LDH is preserved. The TEM image further confirms a well-preserved nanosheet structure with visible layered features (Supplementary Fig. 35b). Strong Ni diffraction peaks are also observed, suggesting that the Ni foam substrate is well protected against chloride corrosion. The Raman spectrum shows the sustained presence of PW_{12} -POM and the formation of metal oxyhydroxides, evidenced by the enhancement of the –OOH peak (Supplementary Fig. 36a). XPS analysis reveals almost no shift or intensity change in the Fe 2p region (Supplementary Fig. 36b), increased Co^{3+} content in the Co 2p region (Supplementary Fig. 36c), and stable W 4f signal intensity (Supplementary Fig. 36d). These results also indicate that PW_{12} -POM stabilizes Fe and promotes the formation of metal oxyhydroxides".

Comment 6: It is strange that, while extensive evidence is presented for the chemical valence and coordination environment of Co and Fe, there is a lack of explanation concerning P and W elements. During operation, especially at ampere-level current densities, would POM decompose? Would the Keggin structure keep stable in alkaline as this unique structure is generally stable in acidic environment? Is the real protective species such as PO_4^{3-} and WO_4^{2-} formed after POM degradation? Therefore, the stability or evolution of POM needs to be addressed, particularly as phosphate has been proven to effectively repel Cl^- . If possible, control experiments using PO_4^{3-} and WO_4^{2-} should be added.

Response: Thank you for this important and constructive suggestion. As correctly noted, our initial manuscript focused primarily on the chemical environment of Co and

Fe, with limited discussion of P and W elements. In the revised version, we have supplemented this analysis. XPS measurements show that the introduction of PW₁₂-POM results in slight positive shifts in the Co 2p region and more pronounced shifts in Fe 2p toward higher binding energy. Concurrently, the W 4f and P 2p signals shift toward lower binding energies, suggesting electron transfer from metal centers to the PW₁₂-POM. However, due to the low phosphorus content in PW₁₂-CoFe LDH (0.34%; Supplementary Table 1) and the inherently low P/W ratio, the XPS signal of P is weak. As a result, we were unable to obtain reliable XANES spectra and post-reaction XPS data for phosphorus. To evaluate POM stability under practical operation, we performed long-term electrolysis at ampere-level current densities. Raman spectra confirm the retention of characteristic PW₁₂-POM vibrational modes after 1300 hours (Fig. R11a), and XPS shows stable W 4f signal intensity (Fig. R11d), indicating that the Keggin structure remains intact in the alkaline seawater environment during ASO.

Comment 7: The citation in Fig. 2e is not comprehensive, the authors should update it, for example 1.25 A cm⁻² for 2800 hours in Ref. 50. The current density of Ref. 50 in Fig. 5c should also be double checked.

Response: We appreciate your comment and thank you for drawing our attention to this oversight. We have corrected the inaccurate presentation of CoFe-Ci@GQDs/NF in the revised manuscript. Specifically, we replaced the cell voltage comparison in Fig. 2e with a tolerance-based analysis, and therefore did not modify the current figure. We fully acknowledge the outstanding performance of CoFe-Ci@GQDs/NF, which achieved approximately 1.25 A cm⁻² for 2800 hours in simulated seawater via CO₃²⁻ intercalation and surface decoration with graphene quantum dots, making it one of the most durable anodes to date. Since our system was tested in real seawater, we maintained the original comparison. However, to ensure a fair and complete comparison, we have included the data from Ref. 50 in Table 3 and revised the corresponding discussion.

Comment 8: The potential configuration of POM on CoFe-LDH should be carefully

discussed, with only one coordination site as the authors proposed in Fig. S27, or more sites are possible. The unique roles of POM in the reaction mechanism, electronic modulation, protection should be intensified.

Response: Thank you for raising this important point. To explore the possible configurations of PW₁₂-POM on CoFe LDH, we constructed two adsorption models: one on a Fe site and another on a Co site (Fig. R12; Supplementary Fig. 29). The DFT-calculated binding energies suggest a clear preference for Fe, with 0.103 eV, compared to 0.864 eV for Co. This is consistent with the electronic structure analysis from XPS and XANES. Although we depict only one coordination site in the model, it is representative of the repeated sites across the LDH surface due to structural symmetry. Furthermore, the revised manuscript has expanded the discussion of the unique functions of PW₁₂-POM, including its roles in promoting active phase evolution, modulating the electronic environment, and protecting against chloride-induced degradation, supported by both in situ XAS and theoretical results.

Fig. R12. Structural diagrams of PW₁₂-CoFe LDH connected via (a) Fe and (b) Co sites.

Point-by-point response to the reviewers #3

Reviewer #3 (Remarks to the Author): This The authors of: 'Engineered polyoxometalate-docked Fe sites on CoFe hydroxide anodes towards industrial-level alkaline seawater electrolysis' have used a variety of different techniques to prove the effect of polyoxometalates on repelling Cl⁻ adsorption and increase at the same time faradaic efficiency towards OER and stability. However, I have a few key questions. In general this is a very solid work that after a few modifications looks like it can have a significant impact on sea water electrolysis by easy synthesis of catalysts and a simple surface modification.

Response: We sincerely thank you for your constructive feedback and positive evaluation of our work. In response to your comments, we have carefully revised the manuscript and addressed each point in detail. We believe these changes have strengthened both the technical content and overall presentation of the study, and we hope it is now suitable for publication.

Comment 1: In Fig. 1j the authors calculate the Fe oxidation state based on the peak of the 1st derivative which obviously is not well smoothed and the authors chose the far left peak for the CoFe-LDH catalyst and far right for the POM-CoFe-LDH, which meant a E₀ of 7124 and 7128 respectively giving huge difference in the expected oxidation state, from 2.62 to 3.51. One can see that with a correct smoothing factor the oxidation states are very similar.

Response: We appreciate your observation. To determine the oxidation state of Fe, we analyzed the first derivative of the XANES spectrum to identify the energy at the steepest slope of the absorption edge. This energy closely corresponds to the electronic transition threshold and is widely used for oxidation state assessment. We deliberately chose not to apply smoothing to the derivative, as it may artificially shift the inflection points and reduce precision in edge energy determination. The extreme points observed in the derivative curves for CoFe-LDH and PW₁₂-CoFe-LDH provide consistent and reproducible indicators of oxidation state differences. Although smoothing can enhance visual clarity, we prioritized analytical accuracy in our

evaluation. These methodological choices are now further clarified in the revised text.

Comment 2: In Fig. 1k the authors suggest a Fe-O-Co/W bond at around 2.5Å, however the same peak can be seen for the FeO and the Fe₂O₃. Why is that?

Response: Thank you for pointing this out. While the EXAFS peak around 2.5 Å does appear in FeO and Fe₂O₃ and corresponds to Fe–O–Fe interactions, the origin of the similar peak in our PW₁₂-CoFe LDH system is different. In CoFe LDH, this peak arises from Fe–O–Co coordination. After PW₁₂-POM coordination, we observe a shift and elongation in this peak, which we attribute to the formation of Fe–O–W bonds due to direct interaction with the oxygen atoms of PW₁₂-POM. Although the bond length is similar, the chemical identity and spatial configuration of the scatterers differ significantly. The replacement of Fe or Co with W in the second coordination shell alters the scattering potential and reflects the new bonding environment.

Comment 3: Additionally, the calculations on ΔGOH for the 2 catalysts also give abnormally different energy values with and without POM. Also OH adsorption by itself does not solely govern catalytic activity.

Response: We appreciate your thoughtful feedback. While our initial discussion emphasized *OH adsorption, we agree that full evaluation of OER kinetics requires a broader analysis. We therefore conducted further calculations to assess the entire OER mechanism on both Co and Fe sites. Although the introduction of PW₁₂-POM lowers the *OH adsorption free energy at Co sites to –0.836 eV, the energy barrier for the rate-limiting step (*O to *OOH) is actually reduced in PW₁₂-CoFeOOH compared to CoOOH (2.425 eV vs. 2.487 eV), as shown in **Fig. R7a** and Supplementary Fig. 39. When Fe is considered as the active site, the situation differs. Fe binds *OH more strongly, but the conversion from *OOH to O₂ becomes highly unfavorable due to an excessive energy barrier (**Fig. R6**). This illustrates a typical Sabatier limitation, where too strong adsorption impedes catalytic turnover. In contrast, Co provides a better balance between binding strength and reaction kinetics, reinforcing its role as the active site. These details have been included in the revised manuscript for clarity.

Response to the reviewers #4

Reviewer #4 (Remarks to the Author): *The authors have developed polyoxometalate (POM) based POM-CoFe LDH/NF anode for possible use in electrolyzers that use saline water with high Cl⁻ content. The POM coating provides stability to the anode by avoiding Cl⁻ mediated metal leaching and enhances dehydrogenation by modulating electron density on active Co-sites. POM-based materials for various electrolyzers have been developed over the years, and I think the authors should have a look at the latest review article, <https://doi.org/10.1039/D3QM01000G>.*

In any case, the work presented here is well executed and detailed studies on each aspect of the electrochemical processes have been carried out. I do not have many comments except a minor query related to the computational part of the work.

All the DFT calculations were carried out in the gas phase at 0 K temperature. Although temperature correction takes care of the TΔS part, how does the model describe solution phase reactions on the electrode surface? Typically the OH⁻ and Cl⁻ ions are well-solvated in a realistic system, and their attachment will be largely affected by a desolvation-type attachment mechanism. The solvent dielectric might impact the charge density distribution significantly. At least, the implicit model description is necessary to incorporate the solvent effect. A cautionary comment on this will be required while comparing experimental and computational outcomes.

Response: Thank you for your constructive comments and positive evaluation of our work. As noted, our DFT calculations were carried out in the gas phase at 0 K, without explicitly accounting for solvent effects such as ion solvation, desolvation barriers, or dielectric screening. These factors can indeed influence the adsorption behavior of OH⁻ and Cl⁻, and may significantly affect charge distribution and interfacial reaction dynamics. To address this limitation, we have added a cautionary note in the "Calculation Methods" section of the revised manuscript, clearly stating that the computed values are approximations and should be interpreted with care when compared to experimental results. Additionally, we thank you for recommending the recent review on polyoxometalate-based materials (DOI: 10.1039/D3QM01000G). We have consulted this article, and its insights have informed the revised discussion.

Reviewer #1 (Remarks to the Author): Most of my concerns are addressed in the revised manuscript. But there are still some problems.

Comment 1: The authors synthesized and characterized CoFe LDH catalysts modified with PW₁₂-POM, SiW₁₂-POM, and PMo₁₂-POM. They declare that “unlike PW₁₂-POM, the incorporation of SiW₁₂-POM or PMo₁₂-POM shifts the Co 2p peaks to lower binding energies, suggesting electron accumulation at Co sites and a reduced oxidation state”. However, the shift of binding energy for Co is unobvious. Moreover, the binding energies of W and Mo for SiW₁₂-POM, and PMo₁₂-POM both shift negatively. How can the Co sites accumulate electrons under such condition? Besides, why does PW₁₂-POM show different performance compared with SiW₁₂-POM and PMo₁₂-POM? How would Si affect the interaction between different metal sites? What is the difference between W and Mo? More in-depth theoretical discussion should be provided to improve the scientific significance of this work.

Response 1: We greatly appreciate your thoughtful comments, which have motivated us to provide a deeper theoretical context and clearer explanations. Following your valuable suggestions, we have carefully revised our discussion and added new charge density difference analyses (Fig. R1, R2; Supplementary Figs. 30, 31) to explain the observed electronic phenomena.

Specifically, our DFT calculations demonstrate that PW₁₂-POM preferentially adsorbs onto Fe sites, displaying a significantly lower binding energy at Fe (0.103 eV) compared to Co (0.864 eV), matching closely with our experimental XPS and XANES data that show substantial electron depletion at Fe sites. In contrast, SiW₁₂-POM and PMo₁₂-POM exhibit comparatively smaller energy differences between Fe and Co sites (SiW₁₂-POM: Fe 0.461 eV, Co 0.776 eV; PMo₁₂-POM: Fe 0.380 eV, Co 0.816 eV). This suggests the existence of competitive adsorption, predominantly at Fe sites but partially also at Co sites. Experimentally, the observed negative shifts in the binding energies of W 4f and Mo 3d upon SiW₁₂-POM and PMo₁₂-POM adsorption indicate electron transfer primarily from Fe to these POM. Charge density difference diagrams further reveal that, while Fe sites serve as the major electron donors, a minor portion of electrons can accumulate toward Co when SiW₁₂-POM or PMo₁₂-POM are adsorbed at Co site. This

back-transfer could account for the subtle electron accumulation at Co, thus explaining the modest negative shifts of Co 2p peaks, thereby reducing catalytic activity. Specifically, the highly electronegative central P⁵⁺ and stable W⁶⁺ centers in PW₁₂-POM may strongly withdraw electrons from metal sites. The weaker acidity of SiW₁₂-POM arising from Si⁴⁺ and the readily reducible Mo⁶⁺ center in PMo₁₂-POM may restrict effective electron extraction, thereby impacting their catalytic activity.

Fig. R1 Charge density difference diagram of PMo₁₂-CoFe LDH connected via (a) Fe and (b) Co sites.

Fig. R2 Charge density difference diagram of SiW₁₂-CoFe LDH connected via (a) Fe and (b) Co sites.

Comment 2: In Figure 5, the operation voltage of the cell at 60 °C is higher than that of 25 °C in the previous version. The author should check the data and provide explanation for the strange performance. Moreover, some recent progresses should be added in Figure 5d, such as *Nature*, 2025, 639, 360.

Response 2: We appreciate your detailed review and the opportunity to clarify this

point. The higher operation voltage observed at 60 °C compared to 25 °C in the previous version is mainly attributed to the use of different recording methods: the polarization curves were measured using a DC power supply, whereas the stability tests were conducted using a Land CT2001A system. In the revision, we followed your suggestion that the MEA should be evaluated at 60–80 °C rather than at room temperature. We have therefore updated the corresponding experimental conditions accordingly. We also appreciate the recommendation to include the recent progress (Nature, 2025, 639, 360). This work represents an important contribution to addressing degradation under intermittent operation for seawater electrolysis, and we have now incorporated it into Figure 5d.

We sincerely thank you for your careful review and constructive remarks. Your suggestions have been immensely helpful in improving the clarity and overall quality of our manuscript.